# TwinQuant: Learnable Subspace Decomposition for 4-Bit LLM Quantization

Haodong Wang [1]   Junjie Liu [2]   Zicong Hong [1]   Qianli Liu [1]   Jian Lin [1]   Song Guo [1]   Xu Chen [2]

## Abstract

4-bit quantization reduces the memory footprint and latency of large language model inference, but its aggressive precision reduction can severely degrade accuracy. Prior methods address this by decomposing each weight matrix into two components (e.g., via singular value decomposition) and quantizing them separately, assigning the bulk of values to a low-precision residual component while handling outliers with a high-precision low-rank component. However, such decompositions are designed to minimize the real-valued energy of the residual, rather than the post-quantization error of the residual and low-rank components. We propose TwinQuant, a 4-bit quantization framework that learns quantization-friendly decomposed subspaces and jointly reshapes both the low-rank and residual components. TwinQuant learns component-specific transformations via a joint optimization over the Stiefel and general linear manifolds, flattening their distributions and reducing dynamic-range imbalance. To enable efficient end-to-end execution, we further design a fused dual-component kernel that pipelines the two-stage low-rank computation on-chip and merges both components with a single epilogue, avoiding intermediate global-memory traffic. Across LLaMA3 and Qwen3 models, TwinQuant preserves near-FP16 accuracy and delivers up to $1.8\times$ end-to-end speedup over an FP16 baseline.

## 1. Introduction

Large language models (LLMs) have become core infrastructure for conversational assistants, medical deci-sion support, and code generation (Grattafiori et al., 2024; Yang et al., 2025; Chen et al., 2026; Team et al., 2023; Thirunavukarasu et al., 2023; Barke et al., 2023), driven largely by aggressive scaling from 7B to 70B and even beyond 600B parameters (Liu et al., 2024). However, this growth makes inference increasingly dominated by large matrix multiplications, amplifying both compute and memory-traffic costs and creating a key deployment bottleneck.

To relieve this pressure, modern accelerators are rapidly strengthening native support for low-precision arithmetic (e.g., FP8/FP4 on NVIDIA Blackwell) (NVIDIA, 2024). Meanwhile, post-training quantization (PTQ) (Ashkboos et al., 2025; Lin et al., 2025; Zhao et al., 2024b) compresses weights and runtime activations into low-bit formats, reducing memory footprint and improving throughput, making quantization a key technique for efficient LLM serving.

However, aggressively quantizing both weights and activations to low-bit often leads to large accuracy drops because their heavy-tailed and highly anisotropic statistics couple heterogeneous components within the same tensors, causing the quantization error to be dominated by a small subset of directions and magnitudes (Wang et al., 2025). Early approaches apply scaling and matrix transformations (Ashkboos et al., 2025; Liu et al., 2025; Sun et al., 2025) to re-distribute weights and activations before quantization, but they mainly re-express tensors in a different coordinate system without removing their intrinsic anisotropy: a few dominant directions still dictate the dynamic range, forcing all groups to share limited 4-bit scales and leaving substantial distortion that redistribution alone cannot eliminate.

More recent methods are to move beyond a single space and decompose each weight into two components (Li et al., 2025; Yao et al., 2024). Representative singular value decomposition (SVD)-based schemes (Li et al., 2025), where a truncated SVD can absorb outliers into a small high-precision low-rank branch while quantizing the residual. This is effective when singular values decay quickly, so a tiny rank can preserve most outliers in diffusion models. However, we empirically observe much slower rank decay in the mainstream LLMs (as discussed in section 3), so tiny rank cannot absorb enough outliers. This creates a dilemma: increasing rank adds substantial memory and compute overhead, while a small rank leaves a large residual with sharp

---

[1]Department of Computer Science and Engineering, The Hong Kong University of Science and Technology, Hong Kong, China. [2]School of Computer Science and Engineering, Sun Yat-sen University, Guangzhou, China. Correspondence to: Zicong Hong <ziconghong@gmail.com>, Song Guo <songguo@ust.hk>.

*Proceedings of the $43^{rd}$ International Conference on Machine Learning*, Seoul, South Korea. PMLR 306, 2026. Copyright 2026 by the author(s).

4-bit accuracy loss.

To break this trade-off, we propose **TwinQuant**, a dual-branch 4-bit quantization framework for LLMs built on a *learnable subspace decomposition* that learns how to split each weight into two complementary components under a limited rank budget, with both components being quantization-friendly. By learning the decomposition, Twin-Quant reshapes the statistics of each component to be more amenable to low-bit quantizers (e.g., reducing dynamic range and directional skew) so that 4-bit distortion is lowered without increasing the truncation rank or introducing substantial extra compute and memory overhead.

A naive strategy is to perform an SVD decomposition and then train the quantization parameters (e.g., quantization scales and clipping thresholds) for the low-rank and residual components. However, this faces two practical issues: (i) tuning these parameters across all layers and both components introduces non-trivial training overhead, often approaching the cost of full quantization-aware training; (ii) the two components are strongly coupled: the residual is defined by what the low-rank reconstruction does not capture, so updating the quantization of one component shifts the effective distribution seen by the other, making joint calibration unstable and difficult to coordinate.

Inspired by rotation-based quantization (Liu et al., 2025; Ashkboos et al., 2025) to reshape tensor statistics for low-bit quantization without heavy training overhead, TwinQuant uses parameter-efficient global and layer-specific transforms to adaptively reallocate the subspaces of the low-rank and residual components, directly minimizing quantization error. In addition, these transforms can be seamlessly folded into the weights offline, avoiding additional overhead at inference time. To further reduce the runtime cost of the two-component computation, we design a specialized kernel that jointly executes both components while minimizing memory-access overhead.

We summarize our contributions as follows:

- We propose **TwinQuant**, a 4-bit quantization paradigm for LLMs that learns quantization-friendly decomposed subspaces and efficiently reallocates information between the low-rank and residual components.

- We introduce a **joint optimization scheme** that over the Stiefel and general linear manifolds to learn better component-specific transformations for two components to minimize end-to-end quantization error.

- We design a **fused dual-component kernel** that pipelines the two-stage low-rank computation on-chip and merges both components in a single epilogue, reducing kernel launches and avoiding intermediate global memory traffic.

- We conduct extensive experiments on mainstream LLMs and system-level evaluations on both workstation and server-class GPUs, demonstrating that Twin-Quant preserves near-FP16 accuracy while achieving up to $1.8\times$ speedup.

## 2. Background & Related Work

Quantization is a widely used technique to accelerate the linear operators in Transformer layers by reducing their arithmetic precision. For a tensor $\mathbf{X}$, symmetric quantization is defined as

$$\mathbf{Q_X} = \text{round}\left(\frac{\mathbf{X}}{s_\mathbf{X}}\right), \quad s_\mathbf{X} = \frac{\max(|\mathbf{X}|)}{q_{\max}}, \qquad (1)$$

where $\mathbf{Q_X}$ is the integer representation of $\mathbf{X}$, $s_\mathbf{X}$ is the scaling factor, and $q_{\max}$ is the maximum representable integer. For signed integers with $k$ bits, $q_{\max} = 2^{k-1} - 1$. The dequantized tensor is then given by $\mathcal{Q}(\mathbf{X}) = s_\mathbf{X} \mathbf{Q_X}$. For a linear layer with activation $\mathbf{X} \in \mathbb{R}^{b \times m}$ and weight $\mathbf{W} \in \mathbb{R}^{m \times n}$,

$$\mathbf{XW} \approx \mathcal{Q}(\mathbf{X})\,\mathcal{Q}(\mathbf{W}) = s_\mathbf{X} s_\mathbf{W} \cdot \mathbf{Q_X}\mathbf{Q_W}. \qquad (2)$$

To exploit low-precision compute units on modern GPUs, $\mathbf{Q_X}$ and $\mathbf{Q_W}$ usually use the same bit-width; otherwise, one must be dequantized on the fly to match the other, which undermines quantization speedups. We also define the quantization error as $\mathbf{E_X} = \mathbf{X} - \mathcal{Q}(\mathbf{X})$.

**Per-channel Scaling.** The input activations are often rich in outliers. To mitigate their impact on quantization, a popular way is to apply channel-wise scaling over weights and activations

$$\hat{\mathbf{X}}\hat{\mathbf{W}} = \left(\mathbf{X}\,\text{diag}(\lambda)^{-1}\right) \cdot \left(\text{diag}(\lambda)\mathbf{W}\right),$$

where $\lambda$ is the channel-wise scaling factor which jointly calibrates the magnitudes of input activations and weights and can be folded into the model weights offline, eliminating extra runtime computation. SmoothQuant (Xiao et al., 2023) selects the scaling factors via grid search to balance the activation–weight trade-off, while FlatQuant (Sun et al., 2025) learns the scaling factors directly as trainable parameters.

**Low-Rank Decomposition.** Recent works show that low-rank decomposition can effectively reduce the errors introduced by low-bit quantization in diffusion models (Li et al., 2025; Yao et al., 2024). SVDQuant (Li et al., 2025) applies SVD to the scaled weights, absorbing most of the outlier components into a low-rank component with controlled overhead, while quantizing the remaining residual component to low precision:

$$\hat{\mathbf{Y}} = \hat{\mathbf{X}}\hat{\mathbf{W}} \approx \mathcal{Q}(\hat{\mathbf{X}})\mathbf{UV} + \mathcal{Q}(\hat{\mathbf{X}})\mathcal{Q}(\mathbf{R}), \qquad (3)$$

where $\mathbf{R} \in \mathbb{R}^{m \times n}$ is the residual component and $\mathbf{U} \in \mathbb{R}^{m \times r}$, $\mathbf{V} \in \mathbb{R}^{r \times n}$ is the low-rank branch and obtained

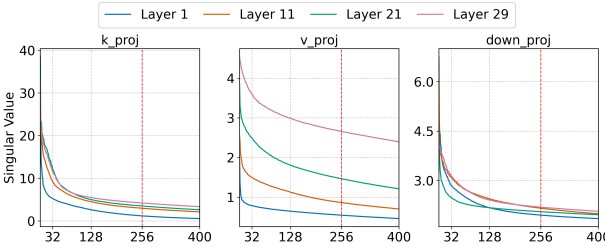

*(a)* First 400 Singular-Value of Selected Linear Layers.

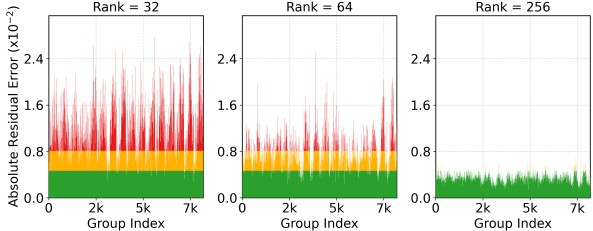

*(b)* Quantization Error Distributions of $\mathbf{R}_k$ in $29^{th}$ layer across different rank numbers. Red denotes large errors, while green denotes small ones.

*Figure 1.* LLM weights exhibit slow spectral decay, leading to rank-sensitive residual quantization error.

by SVD. Empirically, $r \ll \min(m, n)$, thus the additional parameters and computation for the low-rank component are negligible, contributing only $\frac{mr+nr}{mn}$ to the overall costs. However, our goal is to further quantize the low-rank component and learn a quantization-friendly subspace decomposition, rather than relying on a fixed offline SVD.

**Parameter-Efficient Learnable Quantization.** Related work on learnable-parameter LLM quantization keeps pretrained weights fixed but learns a small set of calibration parameters to minimize the quantization error (Zhao et al., 2024a; Sun et al., 2025). For example, Omni-Quant (Shao et al., 2024) learns layer-wise clipping thresholds via short calibration to reduce low-bit distortion, while SpinQuant (Liu et al., 2025) learns rotation matrices that reshape tensor statistics for quantization and can be folded into weights offline. However, unlike these methods that directly calibrate a single quantized weight space, our work learns a decomposed subspace structure tailored to 4-bit quantization.

## 3. Motivation

**Observation 1: LLM Weights Exhibit Slow Singular-Value Decay.** The effectiveness of SVDQuant (Li et al., 2025) relies on a key assumption: the singular-value spectra decay rapidly, so a small low-rank component (e.g. rank=32) can absorb outliers and leave the residual component easier to quantize. This assumption holds well for diffusion models, but we find that it does not directly transfer to main-

stream LLMs. As shown in Figure 1a, the singular-value spectra of LLaMA3-8B decay much more slowly and remain relatively flat even beyond $r \approx 256$. This indicates that many directions still carry non-negligible energy, making the small low-rank setting insufficient for LLM weight decomposition.

**Observation 2: Slow Spectral Decay Creates a Rank–Overhead Dilemma.** This slow singular-value decay directly affects residual quantization. Figure 1b shows the group-wise quantization error distribution of $\mathbf{R}_k$ at the $29^{th}$ layer: rank=32 yields substantial residual errors, while rank=256 significantly reduces them, suggesting that higher ranks absorb more large-magnitude directions. However, the online cost grows roughly linearly with $r$: $\mathbf{UV}$ introduces $r(m + n)$ additional parameters in FP16. For example, the parameters ratio of $q_{proj}$ layer in LLaMA3-8B accounts for $\sim 6.25\%$ of the *4-bit residual-component* storage at $r = 32$, and increases to $\sim 50\%$ at $r = 256$, with proportional additional compute and memory access overhead. Therefore, to make low-rank decomposition practically beneficial, the low-rank component should also be stored and computed in low precision in LLMs.

**Observation 3: Directly Quantizing the Low-Rank Component Introduces Large Error.** The above rank–overhead dilemma suggests that both the low-rank and residual components should be stored and computed in 4-bit. However, we observe that naively quantizing the SVD-based low-rank component to 4-bit can introduce substantial errors, potentially offsetting the benefits of decomposition. Figure 2 reports the group-wise error distribution when we apply SVD (rank=256) to the $q_{proj}$ layer of LLaMA3-8B at three depths and then quantize the low-rank component in 4-bit. This yields a highly skewed error profile: while most groups incur small errors, a small fraction produces disproportionately large distortions, which can dominate the overall low-rank quantization error. A key reason is that the low-rank components are often scale-imbalanced and heavytailed, which 4-bit quantization represents poorly. Inspired by rotation-based quantization (Liu et al., 2025; Ashkboos et al., 2025), a naive method inserts a fixed orthogonal transform (e.g., Hadamard) into the low-rank components to mix values across groups. In practice, this helps little since layer statistics vary widely, so any fixed shared transform cannot simultaneously reduce their 4-bit distortion. These observations motivate a *learnable subspace decomposition* that reshapes the weight distributions and directly minimizes quantization error.

## 4. Method

This section introduces TwinQuant from both algorithmic and systems perspectives, with the key significance being

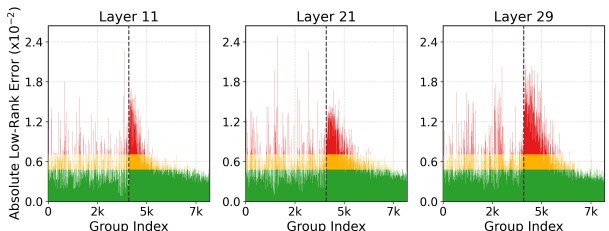

*Figure 2.* Quantization error distributions of low-rank component in three different $q_{\texttt{proj}}$ layers under rank=256.

a *learnable subspace decomposition*, and the overview is shown in Figure 3. We first present a theory-guided re-parameterization over dual decomposed subspaces and derive an error decomposition that motivates learning global orthogonal rotations and layer-specific invertible transforms under structural constraints. We then describe a hybrid manifold optimizer and a fused dual-component kernel for efficient fully 4-bit execution.

### 4.1. Theory-Guided Re-parameterization over Dual Decomposed Subspaces

Our primary objective is to reshape the numerical distributions of the low-rank component, the residual component, and the activations so as to minimize the quantization error:

$$\Delta = \hat{\mathbf{Y}} - \mathcal{Q}(\hat{\mathbf{X}})\left[\mathcal{Q}(\mathbf{U})\,\mathcal{Q}(\mathbf{V}) + \mathcal{Q}(\mathbf{R})\right]. \quad (4)$$

Assuming all quantization noises are mutually independent and independent of the signal, the expected squared error can be rewritten as

$$\mathbb{E}\left[\|\Delta\|_F^2\right] \approx \underbrace{\mathbb{E}\left[\left\|\mathbf{E}_{\hat{\mathbf{X}}}\hat{\mathbf{W}}\right\|_F^2\right]}_{\text{Activation Error}} + \underbrace{\mathbb{E}\left[\left\|\hat{\mathbf{X}}\,\mathbf{E}_{\hat{\mathbf{W}}_{\text{UV}}}\right\|_F^2\right]}_{\text{Low-Rank Weight Error}}$$
$$+ \underbrace{\mathbb{E}\left[\left\|\hat{\mathbf{X}}\,\mathbf{E}_{\mathbf{R}}\right\|_F^2\right]}_{\text{Residual Weight Error}}. \quad (5)$$

where $\mathbf{E}_{\hat{\mathbf{W}}_{\text{UV}}} = \mathbf{E_U}\mathbf{V} + \mathbf{U}\mathbf{E_V}$. A detailed derivation is provided in Appendix A.1. This decomposition shows that the overall error is jointly determined by the quantization errors of activations, low-rank factors, and residual weights, whose magnitudes depend on their value distributions. Therefore, directly quantizing these three parts under a fixed decomposition can be suboptimal, since the decomposed components are not explicitly adjusted to be friendly to 4-bit quantizers.

Recent work (Ashkboos et al., 2025; Liu et al., 2025) has shown that orthogonal matrices (e.g., Hadamard) can re-distribute outliers across channels for both activations and weights, thereby improving low-bit quantization. Twin-Quant uses this observation as the starting intuition, but ex-

tends it from distribution flattening to learnable decomposed subspace re-parameterization. Specifically, we introduce two complementary transforms: the layer-specific invertible matrices $\mathbf{G} \in \mathbb{R}^{r \times r}$ to re-distribute the low-rank component and reduce the low-rank weight error, and the global orthogonal matrices $\mathbf{Q} \in \mathbb{R}^{m \times m}$ to jointly reshape the activation and residual distributions, targeting the remaining two error terms. In this way, $\mathbf{Q}$ and $\mathbf{G}$ do not merely precondition a fixed decomposition, but make the quantized low-rank and residual representation learnable:

$$\mathbf{G}^*, \mathbf{Q}^* = \arg\min_{\mathbf{G},\mathbf{Q}} \left\| \hat{\mathbf{Y}} - \mathcal{Q}(\hat{\mathbf{X}}\mathbf{Q}) \begin{bmatrix} \mathcal{Q}(\mathbf{Q}^{-1}\mathbf{U}\mathbf{G})\,\mathcal{Q}(\mathbf{G}^{-1}\mathbf{V}) \\ + \mathcal{Q}(\mathbf{Q}^{-1}\mathbf{R}) \end{bmatrix} \right\|_F \quad (6)$$

This calibration reconstruction loss is a direct proxy for the quantization-induced output perturbation of the linear layer, as it measures the joint effect of activation, low-rank, and residual quantization errors on the layer output. The transformed weight $\mathbf{U}' = \mathbf{Q}^{-1}\mathbf{U}\mathbf{G}$, $\mathbf{V}' = \mathbf{G}^{-1}\mathbf{V}$ and $\mathbf{R}' = \mathbf{Q}^{-1}\mathbf{R}$ can be pre-processed offline to reduce additional runtime overhead. Although these transformed factors are algebraically equivalent to the original decomposition in full precision, they expose different numerical distributions to the quantizers, which is why optimizing $(\mathbf{Q}, \mathbf{G})$ effectively learns a quantization-aware decomposition.

The following theorem provides theoretical support for our design: it shows that the re-parameterization can provably reduce the expected quantization error, and the improvement is governed by activation flattening gain and low-rank re-parameterization gain.

**Theorem 4.1.** *For any orthogonal transform* $\mathbf{Q}$ *and invertible transform* $\mathbf{G}$, *the re-parameterized subspaces yield an expected quantization error* $\mathbb{E}[\|\Delta'\|_F^2]$ *that satisfies*

$$\mathbb{E}\left[\|\Delta'\|_F^2\right] \leq \frac{\mathbb{E}\left[\|\Delta\|_F^2\right]}{\min\left(\zeta(\mathbf{Q}, \hat{\mathbf{X}}), \eta(\mathbf{G}, \mathbf{Q})\right)}. \quad (7)$$

*Here activation flattening gain* $\zeta(\mathbf{Q}, \hat{\mathbf{X}})$ *and low-rank re-parameterization gain* $\eta(\mathbf{G}, \mathbf{Q})$ *are defined as*

$$\zeta(\mathbf{Q}, \hat{\mathbf{X}}) \triangleq \frac{\mathbb{E}\left[\|\hat{\mathbf{X}}\|_\infty^2\right]}{\mathbb{E}\left[\|\hat{\mathbf{X}}\mathbf{Q}\|_\infty^2\right]},$$
$$\eta(\mathbf{G}, \mathbf{Q}) \triangleq \frac{\mathbb{E}\left[\|\mathbf{U}\|_\infty^2, \|\mathbf{V}\|_F^2 + \|\mathbf{V}\|_\infty^2, \|\mathbf{U}\|_F^2\right]}{\mathbb{E}\left[\|\mathbf{U}'\|_\infty^2, \|\mathbf{V}'\|_F^2 + \|\mathbf{V}'\|_\infty^2, \|\mathbf{U}'\|_F^2\right]}. \quad (8)$$

A detailed proof is provided in Appendix A.1. While Theorem 4.1 motivates searching for optimal $(\mathbf{G}, \mathbf{Q})$, deriving a closed-form solution is generally intractable. This is because the objective involves non-smooth quantization operators and couples $\mathbf{G}$ and $\mathbf{Q}$ through multiple bilinear terms

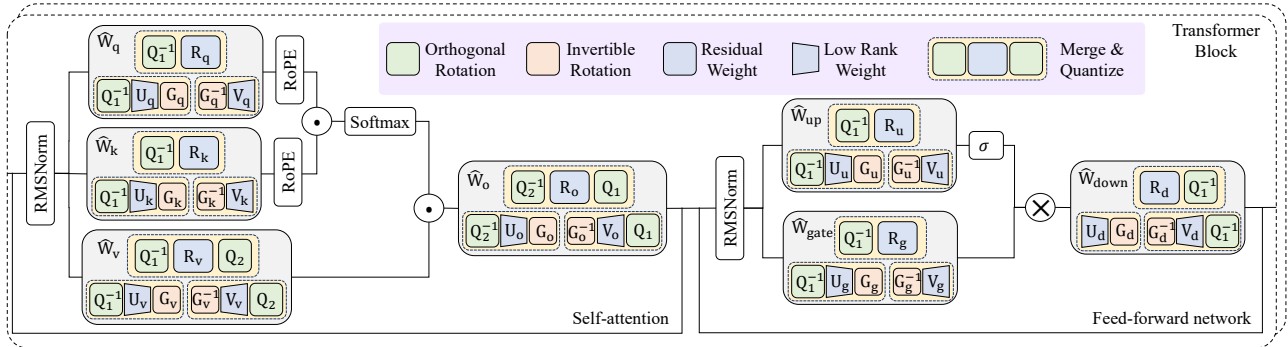

*Figure 3.* The overall framework of TwinQuant. TwinQuant decomposes each linear weight into low-rank and residual components, and learns lightweight global orthogonal and layer-specific invertible transforms to make both branches quantization-friendly. All transforms are folded offline, enabling merged 4-bit inference with no extra runtime overhead.

(e.g., $\mathbf{Q}^{-1}\mathbf{U}\mathbf{G}$ and $\mathbf{G}^{-1}\mathbf{V}$), making the problem highly non-convex even without quantization. Therefore, we adopt a post-training optimization strategy that directly optimizes the transforms on a small calibration set. Specifically, we treat $\mathbf{G}$ and $\mathbf{Q}$ as learnable parameters and minimize the reconstruction error over a few samples to numerically approximate the theoretical optimum. This raises a practical question: how can we optimize these matrices efficiently while respecting their structural constraints?

### 4.2. Joint Optimization on Stiefel and General Linear Manifolds

To further suppress the quantization error while maintaining numerical consistency, we propose a joint optimization framework for the global rotation matrices $\{\mathbf{Q}_1, \mathbf{Q}_2\}$ and the layer-specific transformation $\mathbf{G}$. Unlike previous methods that restrict all transformations to the Stiefel manifold, our approach introduces a hybrid manifold optimization strategy. The optimization objective is formulated as:

$$\mathbf{Q}_1^\star, \mathbf{Q}_2^\star, \mathbf{G}^\star = \underset{\mathbf{Q}_1, \mathbf{Q}_2 \in \mathcal{M},\ \mathbf{G} \in \mathcal{G}}{\arg\min} \mathcal{L}_\mathcal{Q}(\mathbf{Q}_1, \mathbf{Q}_2, \mathbf{G} \mid \hat{\mathbf{W}}, \hat{\mathbf{X}}).$$
(9)

where $\mathcal{M} = \{\mathbf{Q} \in \mathbb{R}^{n \times n} : \mathbf{Q}^\top\mathbf{Q} = \mathbf{I}\}$ denotes the *Stiefel manifold* consisting of all orthogonal matrices and $\mathcal{G} = \mathrm{GL}(n, \mathbb{R})$ denotes the *General Linear manifold* consisting of all invertible matrices. We optimize $\mathbf{G}$ over $\mathcal{G}$ to endow each layer with additional scaling and shearing degrees of freedom ($n^2$ on $\mathcal{G}$ v.s. $\frac{n(n-1)}{2}$ on $\mathcal{M}$) that better absorbs quantization anisotropy and thus reduces error, while $\mathbf{Q}_1$ and $\mathbf{Q}_2$ kept orthogonal on $\mathcal{M}$ since their inverses must be fused across RMSNorm to the adjacent weights, an equivalence that holds for orthogonal transforms but generally breaks for arbitrary invertible transforms, which would otherwise alter the numerical inconsistency.

**Hybrid Manifold Optimizer.** To solve Equation 9, we propose a hybrid optimizer that updates parameters according to their underlying geometry. We keep the global rotations

$\mathbf{Q}_1, \mathbf{Q}_2$ strictly orthonormal for norm preservation, and update them on $\mathcal{M}$ using Cayley SGD (Li et al., 2020):

$$\mathbf{Q}^{(t+1)} = \left(\mathbf{I} - \frac{\alpha}{2}\mathbf{Y}\right)^{-1} \left(\mathbf{I} + \frac{\alpha}{2}\mathbf{Y}\right) \mathbf{Q}^{(t)}, \quad \mathbf{Y} = \hat{\mathbf{O}} - \hat{\mathbf{O}}^\top,$$
(10)

where $\mathbf{O} = \nabla_\mathbf{Q}\mathcal{L}$ is the Euclidean gradient. For the layer-specific transformation $\mathbf{G}$, we optimize them via learned polar parameterization $\mathbf{G} = \mathbf{P}\mathbf{S}$, where $\mathbf{P} \in \mathcal{M}$ is updated by the same Cayley rule and $\mathbf{S}$ is a symmetric positive definite matrix:

$$\mathbf{S} = e^\gamma(\mathbf{L}\mathbf{L}^\top),$$
(11)

with $\mathbf{L}$ a Cholesky factor and $\gamma$ a learnable scale, ensuring $\mathbf{G}$ is always nonsingular and has a well-defined inverse during training. The optimizer performs manifold updates for $(\mathbf{Q}_1, \mathbf{Q}_2, \mathbf{G})$ and momentum SGD for Euclidean parameters $(\mathbf{L}, \gamma)$, while stabilizing training with a conditioning regularizer $\lambda \sum(\|\mathrm{diag}(\mathbf{L})\|^2 + \gamma^2)$ to keep $\mathbf{G}$ near identity at initialization.

**Stage-Wise Decoupling Training.** We implement a three-stage scheduling policy to stabilize the joint optimization: (i) *Global Alignment*: Only $\mathbf{Q}_1, \mathbf{Q}_2$ are trained to align global feature distributions; (ii) *Invertible Adaptation*: Global rotations are frozen, and the invertible $\mathbf{G}$ is optimized to minimize layer-wise quantization error; (iii) *Joint Refinement*: All parameters $\{\mathbf{Q}_1, \mathbf{Q}_2, \mathbf{P}, \mathbf{L}, \gamma\}$ are fine-tuned simultaneously. This schedule is more stable than joint training since the low-rank component is jointly modulated by the global rotations $(\mathbf{Q}_1, \mathbf{Q}_2)$ and its internal transform $(\mathbf{G})$, which makes the optimization highly coupled and prone to gradient conflict. In contrast, our stage-wise decoupling reduces this interference and yields smoother convergence.

### 4.3. Dual-Component Kernel Design

Although the low-rank component has a small theoretical compute cost, executing it as a standalone component can still incur substantial latency overhead. SVDQuant (Li et al.,

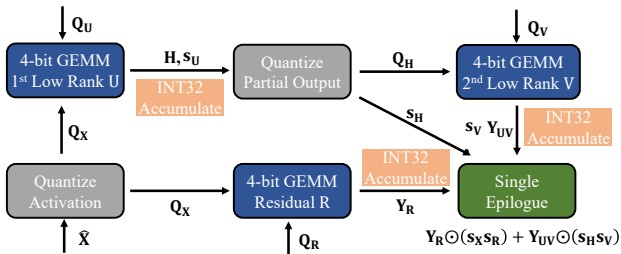

*Figure 4.* The workflow of fused dual-component low-precision kernel. It fuses the low-rank and residual GEMMs with a single epilogue to reduce intermediate memory traffic.

2025) mitigates this issue with a fused operator, but keeps the low-rank path in FP16. In contrast, TwinQuant quantizes *both* the residual and low-rank components to 4-bit, enabling fully low-bit execution while introducing a key systems challenge: the low-rank path is a *two-stage* product, whereas 4-bit TensorCore GEMM can only consume quantized partial outputs. Therefore, the first 4-bit GEMM produces an intermediate $\mathbf{P}$ in INT32 accumulation rather than in an int4-compatible form. Naïvely materializing $\mathbf{P}$ (e.g., INT32), then re-loading and re-quantizing it for the second 4-bit GEMM would incur extra kernel launches and costly global memory traffic, often dominating the low-rank compute. Consequently, $\mathbf{P}$ must be re-quantized and packed *on the fly* and consumed by the second GEMM directly on-chip, without being written back to global memory.

**Fused two-stage low-rank pipeline.** We design a fused kernel that pipelines the two-stage low-rank computation entirely on-chip and merges its output with the residual component before a single write-back. The kernel first quantizes the input activation tile once and reuses it for both components. On the low-rank component, it computes the first 4-bit GEMM in int32 accumulation to obtain an intermediate $\mathbf{H}$, which is kept on-chip and directly prepared as the input to the second 4-bit GEMM. In parallel, the residual component runs a 4-bit GEMM on the same quantized activation. Both component outputs are accumulated in registers and deferred to a shared epilogue for final fusion.

**On-the-fly scaling with a single epilogue.** As shown in Figure 4, we convert the intermediate $\mathbf{H}$ into a packed 4-bit tensor $\mathbf{Q}_H$ in shared memory by estimating an intermediate scale $s_H$ from the INT32 accumulator and re-quantizing on the fly. This eliminates materializing $\mathbf{H}$ in global memory and allows the second low-rank GEMM to directly consume $\mathbf{Q}_H$. Finally, we apply a single epilogue that folds the scale factors and merges the two components, followed by FP16 accumulation and one global store. This single-epilogue design avoids intermediate global-memory traffic for $\mathbf{H}$ and keeps the dual-component execution bandwidth-efficient.

## 5. Experiments

### 5.1. Setups

**Models and metric.** We benchmark against two mainstream LLMs: the LLaMA3 (Grattafiori et al., 2024) models (3B/8B) and Qwen3 (Yang et al., 2025) models (4B/8B/14B/32B). Following previous works (Sun et al., 2025; Liu et al., 2025), we randomly sample the 128 prompts from WikiText2 dataset (Merity et al., 2017) for calibration, each sentence with 2048 tokens. To evaluate the commonsense reasoning capability of our method, we use six zero-shot evaluation tasks, including ARC-Challenge, ARC-Easy (Clark et al., 2018), HellaSwag (Zellers et al., 2019), LAMBADA (Paperno et al., 2016), PIQA (Bisk et al., 2020), and WinoGrande (Sakaguchi et al., 2021). Additionally, we also report the perplexity score on WikiText2 (Merity et al., 2017) datasets. All accuracy metrics are tested on lm-eval (Gao et al., 2024).

**Baselines.** We compare TwinQuant against eight popular PTQ methods, including the naive quantization method RTN (Yao et al., 2023), the weight-only quantization GPTQ (Frantar et al., 2023) and AWQ (Lin et al., 2024), the weight-activation quantization SmoothQuant (Xiao et al., 2023) and three recent state-of-the-art methods QuaRot (Ashkboos et al., 2025), SpinQuant (Liu et al., 2025), FlatQuant (Sun et al., 2025), SVDQuant (Li et al., 2025). Following prior work (Lin et al., 2025; Li et al., 2025), we denote $x$-bit weights and $y$-bit activations as $\mathbf{W}x\mathbf{A}y$. See Appendix A.2 for more details.

**Hardware.** We evaluate TwinQuant on two GPU platforms. *Workstation* (Environment 1): an NVIDIA RTX 4090 (24 GB) GPU paired with an Intel Xeon Gold 6430 CPU. *Inference server* (Environment 2): an NVIDIA L20 (48 GB) GPU paired with an Intel Xeon Platinum 8457C CPU.

**Implementation details.** We apply group-wise quantization to both weights and activations, where each contiguous group of 128 elements shares one quantization scale. We use group size of 128, and rank $r = 128$ as the default setting for TwinQuant. Please refer to Appendix A.6 for more details.

### 5.2. Accuracy Results

Table 1 presents the comparison of WikiText2 perplexity and average accuracy on six zero-shot tasks for LLaMA3 and Qwen3 models. See more details in Appendix A.3.

**Language generation tasks.** Under W4A8, TwinQuant achieves competitive overall perplexity and stays close to full precision. For example, on Qwen3-32B it reduces perplexity from 10.9 (SpinQuant) to 9.9, showing stronger robustness to activation quantization noise. Under the more challenging W4A4 setting, many PTQ baselines exhibit per-

*Table 1.* Comparison of the perplexity score on WikiText2 and averaged accuracy on six zero-shot commonsense reasoning tasks.

| Precision | Method | LLaMA3 3B | | LLaMA3 8B | | Qwen3 4B | | Qwen3 8B | | Qwen3 14B | | Qwen3 32B | |
|---|---|---|---|---|---|---|---|---|---|---|---|---|---|
| | | 0-shot Avg. | Wiki ($\downarrow$) | 0-shot Avg. | Wiki ($\downarrow$) | 0-shot Avg. | Wiki ($\downarrow$) | 0-shot Avg. | Wiki ($\downarrow$) | 0-shot Avg. | Wiki ($\downarrow$) | 0-shot Avg. | Wiki ($\downarrow$) |
| W16A16 | – | 66.2 | 10.7 | 73.5 | 8.6 | 66.8 | 13.7 | 71.6 | 9.7 | 74.2 | 8.6 | 75.2 | 7.6 |
| W4A16 | RTN | 60.9 | 18.8 | 71.2 | 10.5 | 63.3 | 17.6 | 68.1 | 12.0 | 70.3 | 9.9 | 68.1 | 12.7 |
| | GPTQ | 61.7 | 15.2 | 72.4 | 9.0 | 64.3 | 14.5 | 69.7 | 10.3 | 72.7 | 9.2 | 73.5 | 8.3 |
| | AWQ | 63.1 | 12.7 | 72.6 | 10.3 | 64.4 | 16.6 | 72.3 | 10.5 | 69.8 | 9.6 | 73.8 | 8.2 |
| W4A8 | RTN | 60.7 | 29.0 | 71.0 | 10.6 | 58.8 | 30.6 | 64.4 | 12.3 | 68.3 | 11.9 | 67.4 | 13.6 |
| | SmoothQuant | 59.8 | 288.5 | 60.4 | 13.3 | 59.9 | 22.6 | 63.8 | 12.5 | 68.1 | 11.8 | 67.1 | 13.0 |
| | QuaRot | 62.3 | 12.4 | 70.6 | 11.0 | 60.9 | 16.7 | 67.1 | 12.9 | 70.7 | 11.4 | 71.0 | 11.8 |
| | SpinQuant | 64.9 | 11.5 | 72.0 | 9.0 | 65.1 | 19.8 | 69.9 | 12.4 | 72.5 | 11.0 | 73.1 | 10.9 |
| | FlatQuant | 66.9 | 10.7 | 72.5 | 9.8 | 64.4 | 19.4 | 69.9 | 12.0 | 73.4 | 10.6 | 74.4 | 9.8 |
| | TwinQuant | 65.4 | 10.8 | 72.8 | 8.9 | 65.6 | 18.3 | 70.8 | 11.5 | 73.5 | 10.5 | 74.3 | 9.9 |
| W4A4 | RTN | 43.2 | 741.9 | 40.4 | 92.9 | 41.5 | 8791 | 40.1 | 4392 | 42.1 | 18749 | 45.4 | 1796 |
| | SmoothQuant | 44.7 | 372.3 | 52.8 | 19.4 | 40.1 | 9910 | 40.1 | 3360.1 | 41.3 | 21675 | 44.4 | 1806 |
| | QuaRot | 53.4 | 26.9 | 67.4 | 13.1 | 56.6 | 21.5 | 63.4 | 24.5 | 67.2 | 18.2 | 67.4 | 15.6 |
| | SpinQuant | 63.6 | 11.9 | 69.4 | 9.5 | 64.0 | 21.7 | 68.3 | 14.8 | 71.2 | 13.0 | 72.0 | 12.1 |
| | SVDQuant | 64.7 | 11.7 | 68.2 | 12.5 | 63.9 | 20.5 | 68.8 | 14.9 | 71.4 | 12.8 | 72.2 | 11.6 |
| | FlatQuant | 65.5 | 11.3 | 70.1 | 11.0 | 64.1 | 20.0 | 69.3 | 13.4 | 73.1 | 11.4 | 72.8 | 11.0 |
| | TwinQuant | 65.6 | 11.1 | 70.9 | 9.4 | 65.0 | 18.6 | 70.2 | 13.2 | 72.8 | 11.8 | 73.3 | 10.4 |

plexity explosions or collapse (e.g., RTN and SmoothQuant), whereas TwinQuant remains stable across all model families and substantially closes the gap to W16A16. TwinQuant significantly improves over rotation-based QuaRot, reducing perplexity by up to 15.8 points and by 2.9–11.3 points across Qwen3 models. Compared with the affine-based FlatQuant, TwinQuant achieves comparable or better perplexity in most cases, with up to 1.6 points improvement across all models. Compared with SVDQuant (rank=32), TwinQuant further reduces W4A4 perplexity by 0.6–3.1 points across all models.

**Zero-shot tasks.** TwinQuant delivers strong zero-shot accuracy under both W4A8 and W4A4. In W4A8, it is competitive with leading PTQ baselines (e.g., 65.4% on LLaMA3-3B vs. 63.1% for AWQ). In the more challenging W4A4 setting, it performs on par with or better than the best prior methods: on Qwen3-32B it reaches 73.3% average accuracy, exceeding SpinQuant by 1.3 points, FlatQuant by 0.5 points, and SVDQuant by 1.1 points. Across all models, Twin-Quant also consistently outperforms SVDQuant by 0.9–2.7 points in average zero-shot accuracy. While some baselines fail on Qwen and LLaMA models, in contrast, TwinQuant remains stable across both model families, highlighting its robustness under fully 4-bit quantization.

### 5.3. Speed Measurement

Figure 5 summarizes the end-to-end throughput of Twin-Quant over FP16 TensorRT-LLM and recent W4A16 and W4A4 quantization systems with sequence lengths of 1024

input tokens and 256 output tokens in Environments 1 and 2. Compared to the FP16 TensorRT-LLM, TwinQuant achieves 1.63-1.8× and 1.32–1.73× acceleration in Environments 1 and 2, respectively. Against the weight-only baseline AWQ using TensorRT Engine, TwinQuant provides up to 1.74× and 1.6× speedup. When compared with prior SOTA W4A4 methods, TwinQuant delivers an average 2.04× improvement over the rotation-based QuaRot and a 1.59× gain over the affine-based FlatQuant, and further achieves up to 1.07× speedup over the stronger SVDQuant (rank=128). Overall, these results show that TwinQuant consistently outperforms strong baselines, better unlocking the efficiency potential of 4-bit inference on modern GPUs. Additional kernel profile time are reported in Appendix A.4.

### 5.4. Sensitivity Analysis of the Rank Number.

Table 2 and Figure 6 show a clear accuracy–efficiency trade-off with respect to the truncation rank. Increasing rank from 32 to 128 substantially improves model quality: the average zero-shot accuracy increases from 68.0 to 70.9, while Wiki-Text perplexity decreases from 11.3 to 9.4. However, the benefit quickly saturates beyond ($r = 128$). Further increasing rank to 256 only improves the average accuracy by 0.2 point and reduces perplexity by 0.1, but nearly doubles the memory and latency overheads of the low-rank path from 4.9% and 4.3% to 9.9% and 8.7%. These results indicate that larger ranks beyond 128 are not cost-effective, since the extra computation and storage bring negligible quality gains. We therefore use ($r = 128$) as the default setting, which achieves a strong accuracy–overhead trade-off.

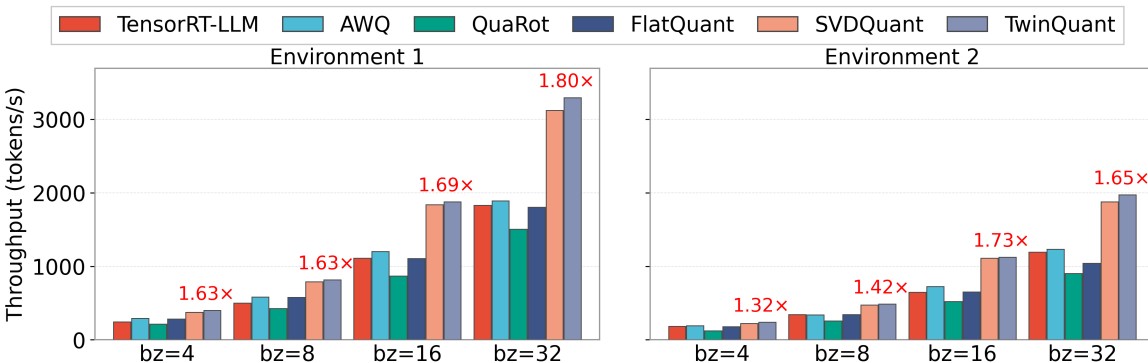

*Figure 5.* Throughput of LLaMA3-8B model across different batch sizes with sequence lengths of 1024 input tokens and 256 output tokens in Environments 1 and 2. SpinQuant is omitted because it does not provide a comparable real W4A4 NVIDIA GPU kernel, making fake-quantized throughput not directly comparable.

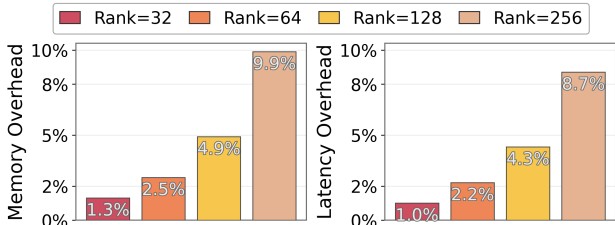

*Figure 6.* Memory and latency overhead of the low-rank component on LLaMA3-8B across different ranks in Environment 1.

| Rank | 0-shot Avg. (%) | Wiki PPL (↓) |
|------|-----------------|--------------|
| 32   | 68.0            | 11.3         |
| 64   | 69.7            | 10.5         |
| 128  | 70.9            | 9.4          |
| 256  | 71.1            | 9.3          |

*Table 2.* Comparison of the average zero-shot accuracy and Wiki-Text perplexity on LLaMA3-8B across different ranks.

## 5.5. Ablation study

As shown in Table 3, we conduct ablation studies on LLaMA3-8B and Qwen3-8B. First, Naive 4-bit directly quantizes the smoothed weights and activations to W4A4 without decomposition, which leads to severe degradation and shows that W4A4 remains highly sensitive to residual outliers and dynamic-range mismatch. Adding an SVD-based low-rank decomposition, where both the low-rank and residual branches are quantized to 4 bits, substantially recovers accuracy. For example, on LLaMA3-8B, the zero-shot average improves from 41.7 to 61.1, and WikiText2 perplexity drops from 91.6 to 19.6. The +Hadamard variant keeps the same decomposed structure but replaces Twin-Quant's learnable transforms with fixed Hadamard transforms, further improves performance (+3.8 and +3.2 points on the zero-shot average). Finally, TwinQuant achieves the

| Models | Method | 0-shot Avg. | Wiki (↓) |
|--------|--------|-------------|----------|
| | FP16 | 73.5 | 8.6 |
| LLaMA3 -8B | Naive 4-bit | 41.7 | 91.6 |
| | +Low-Rank | 61.1 | 19.6 |
| | +Hadamard | 64.9 | 12.4 |
| | TwinQuant | 70.9 | 9.4 |
| | FP16 | 71.6 | 9.7 |
| Qwen3 -8B | Naive 4-bit | 41.5 | 4188 |
| | +Low-Rank | 62.8 | 20.3 |
| | +Hadamard | 66.0 | 16.3 |
| | TwinQuant | 70.2 | 13.2 |

*Table 3.* Ablation study of TwinQuant on LLaMA3-8B and Qwen3-8B, reporting the average accuracy on six zero-shot tasks and WikiText2 perplexity.

best results by using learnable global orthogonal transforms and layer-specific invertible transforms, reaching 70.9 and 70.2 average accuracy as well as 9.4 and 13.2 perplexity on the two models. These results show that TwinQuant benefits from both low-rank decomposition and learnable component-specific transformations.

## 5.6. Quantization Error Analysis

Figure 7 compares the quantization error of the low-rank and residual components under SVD-based decomposition and our learnable subspace decomposition. Across both components, the learnable variant consistently suppresses the long-tail error spikes and reduces the overall error, indicating that it reshapes the parameter distribution into a form that is more amenable to 4-bit quantization. For the low-rank branch, the improvement comes from two sources: a learnable global rotation $\mathbf{Q}$ that aligns the weight space to reduce outlier concentration, and a layer-specific transform $\mathbf{G}$ that adapts the low-rank factors to each layer's statistics, making the rank subspace itself easier to quantize. In contrast, the residual branch does not involve the layer-specific

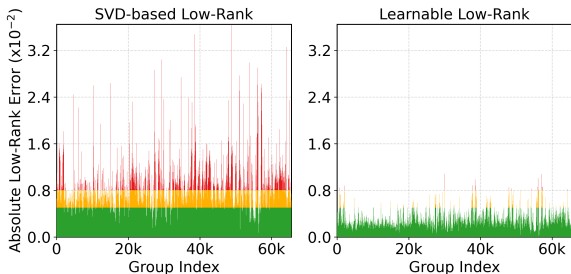

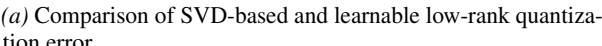

(a) Comparison of SVD-based and learnable low-rank quantization error.

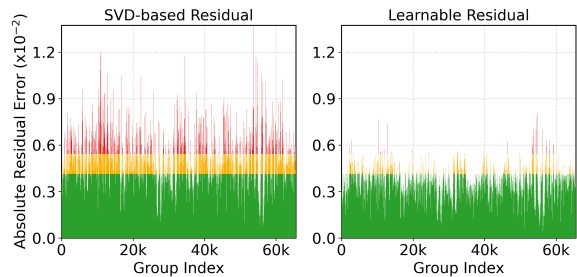

(b) Comparison of SVD-based and learnable residual quantization error.

*Figure 7.* Effect of learnable subspace decomposition on component-wise quantization error in $29^{th}$ layer on LLaMA3-8B. Red denotes large errors, while green denotes small ones.

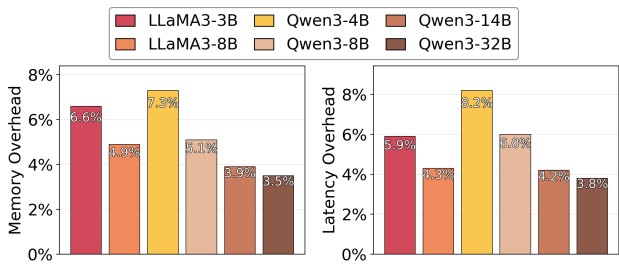

*Figure 8.* Memory and Latency overhead of low-rank component across six models under rank=128 in Environment 1.

**G**; its error reduction mainly stems from the shared global rotation **Q**, which smooths the residual distribution and mitigates the remaining outliers after low-rank extraction. Overall, these results suggest that learning the decomposition subspace provides an effective degree of freedom to jointly lower quantization error in both components.

### 5.7. Overhead of Low-Rank Component

As shown in Figure 8, the low-rank component in Twin-Quant introduces only a limited overhead in both memory and latency across all six models. The memory overhead stays within 3.5–7.3%, as the additional low-rank component is also quantized to 4-bit, so the cost is dominated by a small $r(m + n)$ term rather than storing an extra high-precision branch. We also observe a clear trend that larger models incur smaller overhead (e.g., Qwen3-32B at 3.5% vs. Qwen3-4B at 7.3%), consistent with the fact that a fixed rank becomes relatively cheaper as layer dimensions grow. On the runtime side, the latency overhead remains similarly small (3.8–8.2%), indicating that the extra low-rank computation does not translate into proportional slowdown. This is largely enabled by our dual-component kernel design, which co-schedules the residual and low-rank paths to minimize kernel launches and avoid additional memory traffic, keeping the low-rank update lightweight in the end-to-end inference pipeline.

## 6. Limitations and Future Work

While TwinQuant demonstrates strong accuracy and efficiency on mainstream dense LLMs, several limitations also suggest future directions. First, our current evaluation mainly focuses on dense language models; extending TwinQuant to MoE architectures and multimodal models requires further study, as expert routing and cross-modal modules may introduce different activation and weight distributions. Second, the fused dual-component kernel is optimized for the evaluated GPU platforms, and broader deployment on other accelerator architectures may require additional kernel adaptation and tuning. Third, although our experiments cover representative inference settings, Twin-Quant's effectiveness and system behavior under ultra-long-context workloads require further validation. Finally, the learnable transforms rely on calibration data and introduce additional offline optimization time, motivating future work on more lightweight or data-efficient optimization.

## 7. Conclusion

In this work, we present TwinQuant, a 4-bit PTQ framework that improves low-precision LLM inference throughput without sacrificing accuracy. TwinQuant introduces a learnable subspace decomposition that splits each weight into a low-rank and a residual component, and learns component-specific transformations via a joint optimization scheme over the Stiefel and general linear manifolds to reduce their dynamic range and quantization error. To keep dual-component inference efficient, we develop a customized fused 4-bit kernel that pipelines the two-stage low-rank path entirely on-chip and merges it with the residual path in a single epilogue and write-back. Across mainstream LLMs and GPUs, TwinQuant keeps 4-bit accuracy within 0.6–2.6 average-accuracy points of FP16 with up to $1.8\times$ speedup.

## Impact Statement

This paper presents work whose goal is to advance the field of Machine Learning. There are many potential societal consequences of our work, none of which we feel must be specifically highlighted here.

## Acknowledgments

This research was supported by funding from the Hong Kong RGC General Research Fund (152228/23E, 162161/24E, 162116/25E, 162180/25E), National Natural Science Foundation of China (NSFC) Key Program (No.62532005), Collaborative Research Fund (No. C1042-23GF, No. C5097-25G), NSFC/RGC Collaborative Research Scheme (Grant No. 62461160332 & CRS_HKUST602/24), Research Impact Fund (No. R5011-23F), Areas of Excellence Scheme (AoE/E-601/22-R), National Natural Science Foundation of China (No. U25B2002), Guangdong Basic and Applied Basic Research Foundation (No. 2023B1515120058), and the InnoHK (HK-GAI). We also thank MetaX Integrated Circuits (Shanghai) Co., Ltd for supporting this research.

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

## A. Appendix.

### A.1. Theoretical Analysis

**Derivation of the Expected Quantization Error Decomposition.** For a weight tensor $\mathbf{W}$, the quantization error is defined as $\mathbf{E_W} = \mathbf{W} - \mathcal{Q}(\mathbf{W})$. Then for the low-rank component $\hat{\mathbf{W}}_{\mathrm{UV}} = \mathbf{UV}$, the quantization error can be approximated by neglecting the second-order noise term $\mathbf{E_U}\mathbf{E_V} \approx \mathbf{0}$:

$$\mathbf{E}_{\hat{\mathbf{W}}_{\mathrm{UV}}} \approx \mathcal{Q}(\mathbf{U})\mathcal{Q}(\mathbf{V}) - \mathbf{UV} = \mathbf{E_U}\mathbf{V} + \mathbf{U}\mathbf{E_V}.$$

The total reconstruction error matrix $\boldsymbol{\Delta}$ is defined as the difference between the full-precision output $\hat{\mathbf{Y}} = \hat{\mathbf{X}}(\mathbf{UV} + \mathbf{R})$ and the quantized computation:

$$\begin{aligned}
\boldsymbol{\Delta} &= \hat{\mathbf{Y}} - \mathcal{Q}(\hat{\mathbf{X}})\left[\mathcal{Q}(\mathbf{U})\mathcal{Q}(\mathbf{V}) + \mathcal{Q}(\mathbf{R})\right] \\
&= \hat{\mathbf{X}}\hat{\mathbf{W}} - (\hat{\mathbf{X}} + \mathbf{E}_{\hat{\mathbf{X}}})(\mathbf{UV} + \mathbf{E}_{\hat{\mathbf{W}}_{\mathrm{UV}}} + \mathbf{R} + \mathbf{E_R}) \\
&= \hat{\mathbf{X}}\hat{\mathbf{W}} - (\hat{\mathbf{X}} + \mathbf{E}_{\hat{\mathbf{X}}})(\hat{\mathbf{W}} + \mathbf{E}_{\hat{\mathbf{W}}_{\mathrm{UV}}} + \mathbf{E_R}),
\end{aligned}$$

where $\hat{\mathbf{W}} = \mathbf{UV} + \mathbf{R}$ represents the total weight matrix. By expanding the product and neglecting higher-order noise terms (e.g., $\mathbf{E}_{\hat{\mathbf{X}}}\mathbf{E}_{\mathbf{W}_{\mathrm{UV}}} \approx \mathbf{0}$), we obtain the linearized error:

$$\boldsymbol{\Delta} \approx - \left( \mathbf{E}_{\hat{\mathbf{X}}}\hat{\mathbf{W}} + \hat{\mathbf{X}}\mathbf{E}_{\hat{\mathbf{W}}_{\mathrm{UV}}} + \hat{\mathbf{X}}\mathbf{E_R} \right).$$

Therefore, we have the expected squared Frobenius norm of $\boldsymbol{\Delta}$:

$$\mathbb{E}[\|\Delta\|_F^2] = \mathbb{E}\left[ \left\| \mathbf{E}_{\hat{\mathbf{X}}}\hat{\mathbf{W}} + \hat{\mathbf{X}}\mathbf{E}_{\hat{\mathbf{W}}_{\mathrm{UV}}} + \hat{\mathbf{X}}\mathbf{E_R} \right\|_F^2 \right].$$

Expanding the squared norm of the sum of three matrices yields:

$$\begin{aligned}
\mathbb{E}[\|\Delta\|_F^2] &\approx \mathbb{E}[\|\mathbf{E}_{\hat{\mathbf{X}}}\hat{\mathbf{W}}\|_F^2] + \mathbb{E}[\|\hat{\mathbf{X}}\mathbf{E}_{\hat{\mathbf{W}}_{\mathrm{UV}}}\|_F^2] + \mathbb{E}[\|\hat{\mathbf{X}}\mathbf{E_R}\|_F^2] \\
&\quad + 2\mathbb{E}\left[ \langle \mathbf{E}_{\hat{\mathbf{X}}}\hat{\mathbf{W}}, \hat{\mathbf{X}}\mathbf{E}_{\hat{\mathbf{W}}_{\mathrm{UV}}} \rangle_F + \langle \mathbf{E}_{\hat{\mathbf{X}}}\hat{\mathbf{W}}, \hat{\mathbf{X}}\mathbf{E_R} \rangle_F + \langle \hat{\mathbf{X}}\mathbf{E}_{\hat{\mathbf{W}}_{\mathrm{UV}}}, \hat{\mathbf{X}}\mathbf{E_R} \rangle_F \right].
\end{aligned}$$

Assuming that all quantization noises ($\mathbf{E}_{\hat{\mathbf{X}}}, \mathbf{E_U}, \mathbf{E_V}, \mathbf{E_R}$) are mutually independent and have zero mean ($\mathbb{E}[\mathbf{E}] = \mathbf{0}$), the expectation of the cross-product terms vanishes. For instance:

$$\mathbb{E}\left[ \langle \mathbf{E}_{\hat{\mathbf{X}}}\mathbf{W}, \hat{\mathbf{X}}\mathbf{E_R} \rangle_F \right] = \mathrm{Tr}\left( \mathbf{W}^{\top}\mathbb{E}[\mathbf{E}_{\hat{\mathbf{X}}}]^{\top}\hat{\mathbf{X}}\mathbb{E}[\mathbf{E_R}] \right) = 0. \tag{12}$$

Since all such cross-terms involve at least one noise matrix that is independent of the others, their expectations are all zero. Hence, by retaining only the non-zero terms, the expected squared error is decomposed as:

$$\mathbb{E}\left[\|\Delta\|_F^2\right] \approx \underbrace{\mathbb{E}\left[\left\|\mathbf{E}_{\hat{\mathbf{X}}}\mathbf{W}\right\|_F^2\right]}_{\text{Activation Error}} + \underbrace{\mathbb{E}\left[\left\|\hat{\mathbf{X}}\,\mathbf{E}_{\mathbf{W}_{\mathrm{UV}}}\right\|_F^2\right]}_{\text{Low-Rank Weight Error}} + \underbrace{\mathbb{E}\left[\left\|\hat{\mathbf{X}}\,\mathbf{E_R}\right\|_F^2\right]}_{\text{Residual Weight Error}}. \tag{13}$$

where $\mathbf{E}_{\mathbf{W}_{\mathrm{UV}}} = \mathbf{E_U}\mathbf{V} + \mathbf{U}\mathbf{E_V}$ represents the error contributed by the low-rank factors.

**Proof of Theorem 1.**

*Proof.* For a weight tensor $\mathbf{W}$, the quantization error is defined as $\mathbf{E_W} = \mathbf{W} - \mathcal{Q}(\mathbf{W})$. We assume that the quantization noise $\mathbf{E_A}$ is zero-mean and its variance satisfies $\mathbb{E}[\|\mathbf{E_W}\|_F^2] \propto \mathbb{E}[\|\mathbf{W}\|_\infty^2]$, where the proportionality constant depends on the matrix size and $q_{\mathrm{max}}$. From Equation 4, the expected squared Frobenius norm of a quantized layer is

$$\mathbb{E}\left[\|\Delta\|_F^2\right] \approx \underbrace{\mathbb{E}\left[\left\|\mathbf{E}_{\hat{\mathbf{X}}}\hat{\mathbf{W}}\right\|_F^2\right]}_{\text{Activation Error}} + \underbrace{\mathbb{E}\left[\left\|\hat{\mathbf{X}}\,\mathbf{E}_{\hat{\mathbf{W}}_{\mathrm{UV}}}\right\|_F^2\right]}_{\text{Low-Rank Weight Error}} + \underbrace{\mathbb{E}\left[\left\|\hat{\mathbf{X}}\,\mathbf{E_R}\right\|_F^2\right]}_{\text{Residual Weight Error}} = \mathcal{E}_{act} + \mathcal{E}_{LR} + \mathcal{E}_{res}.$$

where $\hat{\mathbf{W}}_{\mathrm{UV}} = \mathbf{E_U}\mathbf{V} + \mathbf{U}\mathbf{E_V}$. Therefore, in the re-parameterized space, we still have the expected squared Frobenius norm:

$$\mathbb{E}\left[\|\Delta'\|_F^2\right] \approx \underbrace{\mathbb{E}\left[\left\|\mathbf{E}_{\hat{\mathbf{X}}\mathbf{Q}}\hat{\mathbf{W}}\right\|_F^2\right]}_{\text{Activation Error}} + \underbrace{\mathbb{E}\left[\left\|\hat{\mathbf{X}}\mathbf{Q}\,\mathbf{E}_{\hat{\mathbf{W}}_{\mathrm{U'V'}}}\right\|_F^2\right]}_{\text{Low-Rank Weight Error}} + \underbrace{\mathbb{E}\left[\left\|\hat{\mathbf{X}}\mathbf{Q}\,\mathbf{E}_{\mathbf{R'}}\right\|_F^2\right]}_{\text{Residual Weight Error}},$$

where $\mathbf{U}' = \mathbf{Q}^{-1}\mathbf{U}\mathbf{G}$, $\mathbf{V}' = \mathbf{G}^{-1}\mathbf{V}$ and $\mathbf{R}' = \mathbf{Q}^{-1}\mathbf{R}$, then the activation flattening gain $\zeta(\mathbf{Q}, \hat{\mathbf{X}})$ and low-rank re-parameterization gain $\eta(\mathbf{G}, \mathbf{Q})$ can be defined as inverse gains:

$$\zeta(\mathbf{Q}, \hat{\mathbf{X}}) \triangleq \frac{\mathbb{E}\left[\|\hat{\mathbf{X}}\|_\infty^2\right]}{\mathbb{E}\left[\|\hat{\mathbf{X}}\mathbf{Q}\|_\infty^2\right]}, \quad \eta(\mathbf{G}, \mathbf{Q}) \triangleq \frac{\mathbb{E}\left[\|\mathbf{U}\|_\infty^2, \|\mathbf{V}\|_F^2 + \|\mathbf{V}\|_\infty^2, \|\mathbf{U}\|_F^2\right]}{\mathbb{E}\left[\|\mathbf{U}'\|_\infty^2, \|\mathbf{V}'\|_F^2 + \|\mathbf{V}'\|_\infty^2, \|\mathbf{U}'\|_F^2\right]}.$$

For simplicity, we denote $\zeta(\mathbf{Q}, \hat{\mathbf{X}})$ and $\eta(\mathbf{G}, \mathbf{Q})$ by $\zeta$ and $\eta$, respectively. Under these inverse-gain definitions, larger $\zeta$ and $\eta$ indicate stronger reductions in the corresponding quantization-error proxies. Therefore, the re-parameterized activation and low-rank terms satisfy

$$\mathcal{E}'_{act} \leq \frac{\mathcal{E}_{act}}{\zeta}, \qquad \mathcal{E}'_{LR} \leq \frac{\mathcal{E}_{LR}}{\eta}.$$

Since the residual component is also transformed by the same global orthogonal matrix $\mathbf{Q}$, we use the same $\mathbf{Q}$-induced flattening proxy for the residual branch:

$$\mathcal{E}'_{res} \leq \frac{\mathcal{E}_{res}}{\zeta}.$$

Thus, the expected squared error in the re-parameterized space can be bounded as

$$\mathbb{E}\left[\|\Delta'\|_F^2\right] \approx \mathcal{E}'_{act} + \mathcal{E}'_{LR} + \mathcal{E}'_{res} \qquad \leq \frac{\mathcal{E}_{act}}{\zeta} + \frac{\mathcal{E}_{LR}}{\eta} + \frac{\mathcal{E}_{res}}{\zeta} \leq \frac{\mathcal{E}_{act} + \mathcal{E}_{LR} + \mathcal{E}_{res}}{\min(\zeta, \eta)} \qquad \approx \frac{\mathbb{E}\left[\|\Delta\|_F^2\right]}{\min(\zeta, \eta)}.$$

This completes the proof. $\square$

### A.2. Baselines

We benchmark our methods compared with the following eight baselines:

- RTN (Yao et al., 2023) employs uniform rounding-to-nearest with fixed scaling to quantize weights and activations without calibration, thereby offering a almost zero-cost baseline at the expense of larger accuracy loss.

- GPTQ (Frantar et al., 2023) employs post-training, Hessian-aware greedy weight quantization while applying error compensation to previously quantized rows, thereby preserving accuracy under W4A16.

- AWQ (Lin et al., 2024) employs activation-aware weight clipping and per-channel scaling while prioritizing salient channels, thereby achieving accurate W4A16 quantization with light calibration.

- SmoothQuant (Xiao et al., 2023) employs channel-wise scaling to shift activation outliers into weights while smoothing activation distributions, thereby enabling W8A8 quantization with minimal degradation.

- QuaRot (Ashkboos et al., 2025) employs online Hadamard-based orthogonal rotations, implemented with the Fast Hadamard Transform (FHT), to disperse outliers on the fly and quantize in the rotated space, thereby enabling W4A4 quantization with low overhead and small accuracy loss.

- SpinQuant (Liu et al., 2025) employs learned orthogonal rotations to homogenize weight and activation magnitudes while reducing outliers, thereby enabling W4A4 quantization with improved stability.

- SVDQuant (Li et al., 2025) employs an SVD-based low-rank decomposition to isolate outlier-dominated components into an auxiliary low-rank branch and quantize the remaining residual branch, thereby reducing 4-bit quantization error with additional low-rank computation and storage.

- FlatQuant (Sun et al., 2025) employs flattening-based outlier suppression via learnable per-channel affine rescaling to equalize dynamic ranges across channels, thereby reducing quantization error and enabling more accurate low-bit quantization, e.g., W4A4/W4A8, with lightweight calibration.

### A.3. Zero-shot Accuracy

Table 4 reports WikiText2 perplexity and averaged accuracy over six zero-shot benchmarks for LLaMA3-3B and LLaMA3-8B. Under W4A8, TwinQuant remains competitive with the strongest baselines, achieving 65.4/72.8 average accuracy on 3B/8B with modest perplexity increases (10.8 points and 8.9 points, respectively). Notably, several PTQ methods become unstable once activations are quantized: SmoothQuant exhibits severe degradation on 3B (Wiki2 perplexity 288.5), while RTN degrades substantially, indicating sensitivity to activation quantization noise. Under the more challenging W4A4 regime, TwinQuant is consistently robust and achieves the best overall trade-off. On LLaMA3-3B, TwinQuant reaches 65.6 average accuracy, outperforming SpinQuant (63.6 points) and matching FlatQuant (65.5 points), while avoiding the catastrophic failures of RTN/SmoothQuant (perplexity 741.9 and 372.3 points). On LLaMA3-8B, TwinQuant attains 70.9 average accuracy with 9.4 perplexity, improving over SpinQuant (69.4 and 9.5 points) and substantially narrowing the gap to full precision (73.5 and 8.64 points). Overall, TwinQuant delivers stable performance across model scales and quantization settings, especially under fully 4-bit quantization.

Table 5 summarizes WikiText2 perplexity and averaged zero-shot accuracy on four Qwen3 model sizes. Under W4A8, TwinQuant consistently matches or improves upon prior rotation-based baselines and remains among the top-performing PTQ methods. For instance, on Qwen3-4B it achieves 65.6 average accuracy with 18.3 perplexity, improving over SpinQuant (65.1 and 19.8 points) and QuaRot (60.9 and 16.7 points). On larger models, TwinQuant remains competitive (e.g., 74.3 average accuracy on Qwen3-32B) while keeping perplexity in a stable range. Under the harder W4A4 setting, RTN and SmoothQuant largely collapse on Qwen models, exhibiting extremely large perplexity (often in the thousands) and near-random accuracies, indicating severe instability under fully 4-bit quantization. In contrast, TwinQuant remains robust across all scales: it reaches 65.0/70.2/72.8/73.3 average accuracy on 4B/8B/14B/32B, with reasonable perplexity values (18.6/13.2/11.8/10.4). Compared with SpinQuant, TwinQuant improves average accuracy by 1.9/1.3 points on 8B/32B and reduces Wiki2 perplexity notably on 32B (12.1 ($\rightarrow$) 10.4 points). Overall, TwinQuant delivers stable W4A4 performance on Qwen3, consistently narrowing the gap to full precision while avoiding the failure modes of standard PTQ baselines.

### A.4. Kernel Profile

Table 6 shows that kernel fusion consistently reduces both prefill and decode latency across attention and MLP projections on LLaMA3-8B. For attention layers, fusion yields about $1.8$–$2.0\times$ speedup for $q_{\mathrm{proj}}$ and $k/v_{\mathrm{proj}}$ in both phases, and the decode speedup remains close to $2.0\times$ across batch sizes. For MLP layers, fusion provides $1.23$–$1.91\times$ prefill and $1.42$–$2.00\times$ decode improvements, indicating that removing intermediate writes and kernel launches is especially beneficial for decode.

Table 7 confirms similar trends under a different hardware/software setup. Fusion again delivers near $2\times$ improvements for attention projections (e.g., $q_{\mathrm{proj}}$: $1.87$–$2.00\times$ prefill and $1.91$–$1.99\times$ decode; $k/v_{\mathrm{proj}}$: up to $2.23\times$ decode). For MLP projections, prefill speedups are moderate ($1.22$–$1.93\times$), while decode remains consistently accelerated ($1.42$–$2.10\times$). Overall, fusion benefits persist across batch sizes and environments, demonstrating robust system-level gains.

### A.5. Offline Overhead

**Training Overhead.** Table 8 reports the end-to-end optimization cost on $2\times$ NVIDIA RTX Pro 6000. As model size increases, the training time grows predictably (from 27 minutes on LLaMA3-3B to 389 minutes on Qwen3-32B), reflecting the higher compute and memory pressure of larger backbones. In contrast, the quantization stage remains consistently lightweight, taking only 8–38 minutes across all six models and accounting for a small fraction of the overall budget (typically around 10–30%). This indicates that the proposed pipeline introduces modest additional cost beyond standard training, and the offline optimization can be amortized easily in deployment scenarios where models are served for long periods.

*Table 4.* Complete comparison of the perplexity score on WikiText2 and averaged accuracy on six zero-shot tasks on LLaMA3 3B and 8B.

| Model | Precision | Method | ARC-C | ARC-E | HellaSwag | PIQA | Winogrande | LAMBADA | Avg. | Wiki (↓) |
|-------|-----------|--------|-------|-------|-----------|------|------------|---------|------|----------|
| | W16A16 | – | 47.6 | 69.9 | 71.0 | 76.0 | 66.6 | 65.9 | 66.2 | 10.7 |
| | W4A16 | RTN | 41.3 | 60.2 | 66.2 | 73.1 | 63.0 | 61.6 | 60.9 | 18.8 |
| | W4A16 | AWQ | 43.5 | 66.7 | 65.8 | 75.8 | 62.7 | 63.8 | 63.1 | 12.7 |
| | W4A16 | GPTQ | 41.4 | 60.8 | 65.9 | 73.6 | 65.0 | 63.4 | 61.7 | 15.2 |
| | W4A8 | RTN | 42.6 | 60.2 | 66.2 | 72.6 | 62.7 | 60.1 | 60.7 | 29.0 |
| | W4A8 | SpinQuant | 47.2 | 66.8 | 69.2 | 76.0 | 66.7 | 63.2 | 64.9 | 11.5 |
| | W4A8 | QuaRot | 42.9 | 64.3 | 68.1 | 72.6 | 64.8 | 61.2 | 62.3 | 12.4 |
| 3B | W4A8 | SmoothQuant | 40.7 | 59.8 | 65.5 | 73.8 | 58.5 | 60.7 | 59.8 | 288.5 |
| | W4A8 | FlatQuant | 45.1 | 70.7 | 72.2 | 77.0 | 68.1 | 68.2 | 66.9 | 10.7 |
| | W4A8 | TwinQuant | 46.9 | 69.6 | 70.2 | 75.2 | 65.9 | 64.8 | 65.4 | 10.8 |
| | W4A4 | RTN | 29.8 | 41.0 | 41.4 | 57.3 | 50.9 | 38.9 | 43.2 | 741.9 |
| | W4A4 | SpinQuant | 43.9 | 66.3 | 67.2 | 75.0 | 65.5 | 63.5 | 63.6 | 11.9 |
| | W4A4 | QuaRot | 40.6 | 49.9 | 51.2 | 61.9 | 61.4 | 55.2 | 53.4 | 26.9 |
| | W4A4 | SmoothQuant | 30.5 | 43.6 | 37.7 | 58.0 | 52.9 | 45.3 | 44.7 | 372.3 |
| | W4A4 | FlatQuant | 41.9 | 69.4 | 70.8 | 76.1 | 66.4 | 68.2 | 65.5 | 11.3 |
| | W4A4 | SVDQuant | 44.3 | 68.5 | 70.1 | 75.3 | 64.7 | 65.0 | 64.7 | 11.7 |
| | W4A4 | TwinQuant | 45.2 | 69.8 | 70.9 | 76.2 | 66.5 | 65.1 | 65.6 | 11.1 |
| | W16A16 | – | 54.86 | 79.55 | 79.13 | 80.74 | 73.72 | 72.93 | 73.5 | 8.64 |
| | W4A16 | RTN | 52.5 | 77.7 | 77.4 | 79.6 | 73.4 | 66.7 | 71.2 | 10.5 |
| | W4A16 | AWQ | 53.8 | 77.6 | 78.8 | 80.0 | 73.2 | 72.1 | 72.6 | 10.3 |
| | W4A16 | GPTQ | 54.2 | 79.4 | 78.1 | 80.1 | 72.1 | 70.6 | 72.4 | 9.0 |
| | W4A8 | RTN | 52.7 | 77.0 | 77.3 | 79.4 | 73.0 | 66.3 | 71.0 | 10.6 |
| | W4A8 | SpinQuant | 54.0 | 76.5 | 78.1 | 79.6 | 72.4 | 71.3 | 72.0 | 9.0 |
| | W4A8 | QuaRot | 52.4 | 77.0 | 77.1 | 79.7 | 71.0 | 66.5 | 70.6 | 11.0 |
| 8B | W4A8 | SmoothQuant | 42.5 | 66.2 | 69.6 | 73.72 | 66.1 | 57.5 | 60.4 | 13.3 |
| | W4A8 | FlatQuant | 54.2 | 78.3 | 77.5 | 80.2 | 72.4 | 72.3 | 72.5 | 9.8 |
| | W4A8 | TwinQuant | 54.3 | 79.3 | 78.6 | 79.4 | 72.1 | 72.8 | 72.8 | 8.9 |
| | W4A4 | RTN | 28.3 | 40.2 | 43.8 | 59.1 | 50.0 | 21.0 | 40.4 | 92.9 |
| | W4A4 | SpinQuant | 50.9 | 75.0 | 75.9 | 77.5 | 68.5 | 68.4 | 69.4 | 9.5 |
| | W4A4 | QuaRot | 49.2 | 73.4 | 74.1 | 77.4 | 68.3 | 61.8 | 67.4 | 13.1 |
| | W4A4 | SmoothQuant | 35.0 | 57.3 | 59.8 | 67.9 | 56.8 | 40.2 | 52.8 | 19.4 |
| | W4A4 | FlatQuant | 51.2 | 75.3 | 75.6 | 78.3 | 69.9 | 70.1 | 70.1 | 11.0 |
| | W4A4 | SVDQuant | 50.3 | 74.1 | 74.0 | 76.2 | 67.5 | 67.3 | 68.2 | 12.5 |
| | W4A4 | TwinQuant | 52.73 | 76.01 | 76.25 | 78.78 | 71.5 | 69.9 | 70.9 | 9.4 |

## A.6. Implement Detail

We implement the fused dual-component kernel in CUDA/C++ by extending CUTLASS INT4 Tensor Core GEMM templates, requiring roughly 2K lines of additional code. Our training pipeline is built on PyTorch with the HuggingFace Accelerate distributed stack; we further implement a custom trainer (about 4K lines) to support a three-stage schedule and stage-wise learning-rate scaling. Moreover, we train for 1,000 steps in total, consisting of 400 steps of Global Alignment, 400 steps of Invertible Adaptation, and 200 steps of Joint Refinement. We use a constant learning-rate schedule with a base learning rate=5e-3. The effective learning rate is adjusted across stages by scaling the learning rate for the currently optimized parameter subset while freezing the others. We set weight decay=1e-3 globally, but disable weight decay for the rotation-related parameters. Moreover, the smoothing factor $\boldsymbol{\lambda} \in \mathbb{R}^m$ is a per-channel vector whose $i$-th element is computed as $\lambda_i = \max(|\mathbf{X}_{:,i}|)^{\alpha} / \max(|\mathbf{W}_{i,:}|)^{1-\alpha}$ where $\mathbf{X} \in \mathbb{R}^{t \times m}$ and $\mathbf{W} \in \mathbb{R}^{m \times n}$ following SmoothQuant (Xiao et al., 2023). The migration strength $\alpha$ is chosen offline, per layer, by searching for the value that minimizes the layer output mean squared error (MSE) after TwinQuant on the calibration dataset.

*Table 5.* Complete comparison of the perplexity score on WikiText2 and averaged accuracy on six zero-shot tasks on Qwen3 4B/8B/14B/32B.

| Model | Precision | Method | ARC-C | ARC-E | HellaSwag | PIQA | Winogrande | LAMBADA | Avg. | Wiki (↓) |
|---|---|---|---|---|---|---|---|---|---|---|
| 4B | W16A16 | – | 50.6 | 80.5 | 69.5 | 75.0 | 65.8 | 59.4 | 66.8 | 13.7 |
| | W4A16 | RTN | 44.7 | 75.3 | 66.8 | 73.2 | 61.8 | 57.9 | 63.3 | 17.6 |
| | W4A16 | AWQ | 46.2 | 76.7 | 67.8 | 73.8 | 63.1 | 58.5 | 64.4 | 16.6 |
| | W4A16 | GPTQ | 45.4 | 76.6 | 67.0 | 74.4 | 63.9 | 58.2 | 64.3 | 14.5 |
| | W4A8 | RTN | 42.3 | 68.7 | 64.1 | 68.8 | 53.9 | 55.1 | 58.8 | 30.6 |
| | W4A8 | SpinQuant | 49.2 | 78.4 | 67.2 | 73.5 | 63.6 | 58.4 | 65.1 | 19.8 |
| | W4A8 | QuaRot | 43.3 | 70.5 | 66.3 | 71.7 | 57.9 | 55.4 | 60.9 | 16.7 |
| | W4A8 | SmoothQuant | 42.2 | 69.5 | 63.4 | 71.5 | 57.5 | 55.0 | 59.9 | 22.6 |
| | W4A8 | FlatQuant | 50.9 | 73.6 | 66.8 | 74.3 | 63.3 | 57.6 | 64.4 | 19.4 |
| | W4A8 | TwinQuant | 50.2 | 78.9 | 66.9 | 74.5 | 64.2 | 58.6 | 65.6 | 18.3 |
| | W4A4 | RTN | 26.7 | 28.6 | 44.2 | 48.9 | 51.8 | 48.9 | 41.5 | 8791 |
| | W4A4 | SpinQuant | 47.4 | 76.8 | 66.5 | 72.8 | 63.5 | 56.7 | 64.0 | 21.7 |
| | W4A4 | QuaRot | 39.8 | 65.7 | 64.4 | 62.5 | 54.6 | 52.4 | 56.6 | 21.5 |
| | W4A4 | SmoothQuant | 22.6 | 25.9 | 42.3 | 51.6 | 47.9 | 50.2 | 40.1 | 9910 |
| | W4A4 | FlatQuant | 49.4 | 75.3 | 65.5 | 74.0 | 64.0 | 56.2 | 64.1 | 20.0 |
| | W4A4 | SVDQuant | 49.2 | 75.5 | 66.2 | 73.3 | 63.7 | 55.2 | 63.9 | 20.5 |
| | W4A4 | TwinQuant | 49.4 | 77.8 | 67.0 | 73.9 | 64.7 | 57.0 | 65.0 | 18.6 |
| 8B | W16A16 | – | 55.5 | 83.5 | 78.8 | 76.4 | 68.0 | 67.4 | 71.6 | 9.71 |
| | W4A16 | RTN | 53.8 | 79.1 | 76.3 | 75.4 | 63.6 | 60.6 | 68.1 | 12.0 |
| | W4A16 | AWQ | 59.7 | 83.1 | 77.1 | 79.4 | 72.0 | 62.7 | 72.3 | 10.5 |
| | W4A16 | GPTQ | 55.6 | 80.1 | 77.8 | 76.0 | 66.1 | 62.8 | 69.7 | 10.3 |
| | W4A8 | RTN | 45.8 | 73.4 | 72.3 | 73.1 | 62.9 | 58.7 | 64.4 | 12.3 |
| | W4A8 | SpinQuant | 54.2 | 81.9 | 75.5 | 75.6 | 66.5 | 65.8 | 69.9 | 12.4 |
| | W4A8 | QuaRot | 53.9 | 77.8 | 74.8 | 72.7 | 62.6 | 60.5 | 67.1 | 12.9 |
| | W4A8 | SmoothQuant | 44.3 | 71.0 | 71.7 | 73.4 | 62.1 | 60.0 | 63.8 | 12.5 |
| | W4A8 | FlatQuant | 56.8 | 80.6 | 74.2 | 77.1 | 67.3 | 63.1 | 69.9 | 12.0 |
| | W4A8 | TwinQuant | 54.6 | 82.5 | 77.7 | 76.2 | 67.9 | 66.1 | 70.8 | 11.5 |
| | W4A4 | RTN | 22.6 | 24.9 | 45.6 | 48.9 | 51.8 | 46.9 | 40.1 | 4392 |
| | W4A4 | SpinQuant | 53.6 | 78.5 | 71.4 | 76.5 | 67.3 | 62.4 | 68.3 | 14.8 |
| | W4A4 | QuaRot | 49.8 | 74.8 | 69.8 | 68.3 | 60.7 | 57.1 | 63.4 | 24.5 |
| | W4A4 | SmoothQuant | 25.7 | 25.5 | 41.7 | 50.5 | 52.2 | 44.9 | 40.1 | 3360.1 |
| | W4A4 | FlatQuant | 54.4 | 79.4 | 73.1 | 77.0 | 68.8 | 63.4 | 69.3 | 13.4 |
| | W4A4 | SVDQuant | 52.8 | 78.9 | 74.4 | 75.8 | 66.6 | 64.1 | 68.8 | 14.9 |
| | W4A4 | TwinQuant | 53.6 | 80.8 | 77.1 | 75.7 | 67.9 | 65.8 | 70.2 | 13.2 |
| 14B | W16A16 | – | 59.0 | 84.3 | 80.5 | 80.0 | 72.9 | 68.4 | 74.2 | 8.6 |
| | W4A16 | RTN | 51.8 | 80.6 | 78.5 | 77.9 | 68.7 | 64.4 | 70.3 | 9.9 |
| | W4A16 | AWQ | 51.7 | 79.2 | 79.3 | 76.0 | 66.1 | 66.5 | 69.8 | 9.6 |
| | W4A16 | GPTQ | 57.1 | 81.9 | 78.8 | 78.8 | 72.5 | 66.9 | 72.7 | 9.2 |
| | W4A8 | RTN | 50.7 | 79.5 | 75.4 | 75.3 | 66.9 | 62.0 | 68.3 | 11.9 |
| | W4A8 | SpinQuant | 57.8 | 81.8 | 77.6 | 78.3 | 71.8 | 67.6 | 72.5 | 11.0 |
| | W4A8 | QuaRot | 56.4 | 80.7 | 76.1 | 76.9 | 68.5 | 65.8 | 70.7 | 11.4 |
| | W4A8 | SmoothQuant | 50.5 | 79.0 | 75.8 | 75.8 | 66.3 | 61.3 | 68.1 | 11.8 |
| | W4A8 | FlatQuant | 59.3 | 82.0 | 78.1 | 79.8 | 73.4 | 67.6 | 73.4 | 10.6 |
| | W4A8 | TwinQuant | 58.4 | 83.0 | 79.5 | 79.4 | 72.4 | 68.0 | 73.5 | 10.5 |
| | W4A4 | RTN | 24.8 | 23.9 | 50.6 | 55.7 | 46.8 | 50.7 | 42.1 | 18749 |
| | W4A4 | SpinQuant | 56.3 | 81.0 | 75.8 | 76.9 | 71.4 | 65.7 | 71.2 | 13.0 |
| | W4A4 | QuaRot | 52.8 | 76.4 | 72.4 | 73.6 | 65.0 | 62.8 | 67.2 | 18.2 |
| | W4A4 | SmoothQuant | 26.5 | 25.8 | 48.7 | 51.2 | 50.2 | 45.3 | 41.3 | 21675 |
| | W4A4 | FlatQuant | 60.4 | 81.6 | 77.6 | 79.6 | 72.6 | 66.9 | 73.1 | 11.4 |
| | W4A4 | SVDQuant | 56.8 | 80.4 | 76.7 | 78.4 | 71.4 | 64.9 | 71.4 | 12.8 |
| | W4A4 | TwinQuant | 58.0 | 82.1 | 78.6 | 79.0 | 72.8 | 66.1 | 72.8 | 11.8 |
| 32B | W16A16 | – | 57.8 | 84.4 | 84.2 | 80.9 | 73.6 | 70.3 | 75.2 | 7.6 |
| | W4A16 | RTN | 49.7 | 73.0 | 82.6 | 71.9 | 62.9 | 68.5 | 68.1 | 12.7 |
| | W4A16 | AWQ | 56.9 | 82.5 | 83.2 | 79.8 | 71.8 | 68.7 | 73.8 | 8.2 |
| | W4A16 | GPTQ | 57.8 | 82.6 | 83.6 | 79.7 | 67.6 | 69.8 | 73.5 | 8.3 |
| | W4A8 | RTN | 49.5 | 76.3 | 78.6 | 72.9 | 65.3 | 62.0 | 67.4 | 13.6 |
| | W4A8 | SpinQuant | 56.6 | 82.9 | 80.4 | 79.4 | 72.0 | 67.1 | 73.1 | 10.9 |
| | W4A8 | QuaRot | 51.3 | 78.9 | 79.8 | 76.8 | 70.9 | 68.5 | 71.0 | 11.8 |
| | W4A8 | SmoothQuant | 49.2 | 75.8 | 78.9 | 71.4 | 63.5 | 63.7 | 67.1 | 13.0 |
| | W4A8 | FlatQuant | 58.7 | 82.9 | 82.2 | 81.6 | 73.0 | 67.8 | 74.4 | 9.8 |
| | W4A8 | TwinQuant | 56.8 | 82.9 | 83.5 | 80.6 | 73.2 | 68.9 | 74.3 | 9.9 |
| | W4A4 | RTN | 28.5 | 27.6 | 60.8 | 52.5 | 50.7 | 52.0 | 45.4 | 1796 |
| | W4A4 | SpinQuant | 56.7 | 80.6 | 79.2 | 78.9 | 71.8 | 64.8 | 72.0 | 12.1 |
| | W4A4 | QuaRot | 49.8 | 72.2 | 75.6 | 73.4 | 67.9 | 65.3 | 67.4 | 15.6 |
| | W4A4 | SmoothQuant | 26.0 | 28.9 | 56.7 | 51.9 | 54.1 | 48.6 | 44.4 | 1806 |
| | W4A4 | FlatQuant | 57.5 | 81.2 | 80.6 | 80.2 | 71.5 | 65.9 | 72.8 | 11.0 |
| | W4A4 | SVDQuant | 56.1 | 80.9 | 81.1 | 78.6 | 70.8 | 65.6 | 72.2 | 11.6 |
| | W4A4 | TwinQuant | 57.3 | 82.4 | 82.6 | 79.8 | 72.5 | 65.2 | 73.3 | 10.4 |

| Layer Name | Batch Size | without Kernel Fusion | | with Kernel Fusion | | Speedup | |
|---|---|---|---|---|---|---|---|
| | | Prefill Time (ms) | Decode Time (ms) | Prefill Time (ms) | Decode Time (ms) | Prefill | Decode |
| $q_{\text{proj}}$ $4096 \times 4096$ | 1 | 0.214 | 0.226 | 0.116 | 0.113 | 1.84$\times$ | 2.00$\times$ |
| | 2 | 0.227 | 0.228 | 0.124 | 0.113 | 1.83$\times$ | 2.02$\times$ |
| | 4 | 0.295 | 0.228 | 0.173 | 0.112 | 1.71$\times$ | 2.04$\times$ |
| | 8 | 0.511 | 0.224 | 0.318 | 0.113 | 1.61$\times$ | 1.98$\times$ |
| $k_{\text{proj}}$, $v_{\text{proj}}$ $4096 \times 1024$ | 1 | 0.214 | 0.213 | 0.112 | 0.112 | 1.91$\times$ | 1.90$\times$ |
| | 2 | 0.217 | 0.218 | 0.114 | 0.112 | 1.90$\times$ | 1.95$\times$ |
| | 4 | 0.217 | 0.218 | 0.115 | 0.112 | 1.89$\times$ | 1.95$\times$ |
| | 8 | 0.238 | 0.218 | 0.127 | 0.112 | 1.87$\times$ | 1.95$\times$ |
| $up_{\text{proj}}$, $gate_{\text{proj}}$ $4096 \times 14336$ | 1 | 0.325 | 0.215 | 0.170 | 0.116 | 1.91$\times$ | 1.85$\times$ |
| | 2 | 0.394 | 0.223 | 0.269 | 0.117 | 1.46$\times$ | 1.91$\times$ |
| | 4 | 0.601 | 0.226 | 0.452 | 0.114 | 1.33$\times$ | 1.98$\times$ |
| | 8 | 0.987 | 0.226 | 0.801 | 0.113 | 1.23$\times$ | 2.00$\times$ |
| $down_{\text{proj}}$ $14336 \times 4096$ | 1 | 0.295 | 0.301 | 0.210 | 0.212 | 1.40$\times$ | 1.42$\times$ |
| | 2 | 0.323 | 0.303 | 0.247 | 0.201 | 1.31$\times$ | 1.51$\times$ |
| | 4 | 0.632 | 0.302 | 0.488 | 0.200 | 1.30$\times$ | 1.51$\times$ |
| | 8 | 1.167 | 0.306 | 0.907 | 0.209 | 1.29$\times$ | 1.46$\times$ |

*Table 6.* Latency comparison with/without kernel fusion on LLaMA3-8B in Environment 1.

| Layer Name | Batch Size | without Kernel Fusion | | with Kernel Fusion | | Speedup | |
|---|---|---|---|---|---|---|---|
| | | Prefill Time (ms) | Decode Time (ms) | Prefill Time (ms) | Decode Time (ms) | Prefill | Decode |
| $q_{\text{proj}}$ $4096 \times 4096$ | 1 | 0.238 | 0.229 | 0.122 | 0.120 | 1.95$\times$ | 1.91$\times$ |
| | 2 | 0.375 | 0.238 | 0.187 | 0.120 | 2.00$\times$ | 1.99$\times$ |
| | 4 | 0.479 | 0.234 | 0.254 | 0.120 | 1.89$\times$ | 1.95$\times$ |
| | 8 | 0.875 | 0.234 | 0.467 | 0.120 | 1.87$\times$ | 1.95$\times$ |
| $k_{\text{proj}}$, $v_{\text{proj}}$ $4096 \times 1024$ | 1 | 0.222 | 0.265 | 0.119 | 0.119 | 1.86$\times$ | 2.23$\times$ |
| | 2 | 0.223 | 0.252 | 0.119 | 0.119 | 1.87$\times$ | 2.12$\times$ |
| | 4 | 0.214 | 0.253 | 0.122 | 0.119 | 1.76$\times$ | 2.13$\times$ |
| | 8 | 0.378 | 0.237 | 0.207 | 0.119 | 1.83$\times$ | 1.99$\times$ |
| $up_{\text{proj}}$, $gate_{\text{proj}}$ $4096 \times 14336$ | 1 | 0.489 | 0.339 | 0.253 | 0.186 | 1.93$\times$ | 1.82$\times$ |
| | 2 | 0.551 | 0.362 | 0.384 | 0.186 | 1.43$\times$ | 1.95$\times$ |
| | 4 | 0.976 | 0.365 | 0.745 | 0.186 | 1.31$\times$ | 1.96$\times$ |
| | 8 | 1.715 | 0.391 | 1.403 | 0.186 | 1.22$\times$ | 2.10$\times$ |
| $down_{\text{proj}}$ $14336 \times 4096$ | 1 | 0.372 | 0.392 | 0.277 | 0.276 | 1.34$\times$ | 1.42$\times$ |
| | 2 | 0.644 | 0.398 | 0.508 | 0.275 | 1.27$\times$ | 1.45$\times$ |
| | 4 | 1.012 | 0.410 | 0.809 | 0.275 | 1.25$\times$ | 1.49$\times$ |
| | 8 | 1.970 | 0.396 | 1.593 | 0.276 | 1.24$\times$ | 1.43$\times$ |

*Table 7.* Latency comparison with/without kernel fusion on LLaMA3-8B in Environment 2.

| Models | LLaMA3-3B | LLaMA3-8B | Qwen3-4B | Qwen3-8B | Qwen3-14B | Qwen3-32B |
|---|---|---|---|---|---|---|
| Training Time (mins) | 27 | 66 | 58 | 92 | 181 | 389 |
| Quantization Time (mins) | 8 | 14 | 9 | 15 | 25 | 38 |

*Table 8.* Optimization time for six models on 2$\times$ NVIDIA RTX Pro 6000.

