# OpenReview forum: "TwinQuant: Learnable Subspace Decomposition for 4-Bit LLM Quantization"
_ICML.cc/2026/Conference — ICML 2026 regular_

### Official Review · Reviewer_Fq6M · 2026-03-04

**Soundness:** 3
**Presentation:** 3
**Significance:** 3
**Originality:** 3
**Overall Recommendation:** 4
**Confidence:** 4

**Summary:**

This paper introduces TwinQuant, a 4-bit quantization framework for large language models (LLMs). TwinQuant learns quantization-friendly subspaces by jointly optimizing low-rank and residual components. It applies component-specific transformations to reduce distribution imbalance and dynamic range issues, making 4-bit quantization more accurate. The method also includes a fused dual-component kernel for efficient execution.

**Compliance With Llm Reviewing Policy:**

Affirmed.

**Final Justification:**

The paper presents a technically solid and well-motivated approach for low-bit quantization, with a coherent design and strong empirical performance. I believe the method is sound and likely to be useful to the community.

My main concern is the fairness of the experimental setup. In the original submission, different methods use different calibration datasets, which makes the comparison difficult to interpret. Ideally, all results should be reported under a consistent calibration setting.

The rebuttal partially addresses this by providing additional experiments under aligned calibration, which helps clarify the issue.

Overall, I maintain a **weak accept** recommendation.

**Key Questions For Authors:**

Check above

**Limitations:**

Yes

**Strengths And Weaknesses:**

## Strengths

- The motivation and overall methodology are reasonable and well-grounded. The paper combines rotation-based transformations, quantization, and low-rank decomposition in a coherent framework.
- The mathematical formulation is generally sound, and the optimization design is sensible. In particular, the choice of Cayley SGD for optimization on the Stiefel manifold is appropriate and well justified.

## Weaknesses / Questions

1. In the *Qwen3 8B* column of Table 1, it appears that as PPL increases, the 0-shot accuracy also increases. This trend is counterintuitive, since higher PPL usually indicates worse language modeling performance. Could the authors explain why this occurs?

2. SpinQuant is not included in Figure 5. Table 1 suggests that it achieves comparable perplexity.

3. Line 346 states that *“Under W4A8, TwinQuant achieves the best overall perplexity and stays close to full precision.”* However, the table does not provide a summary metric or aggregated score across datasets/tasks. Could the authors clarify what “overall” refers to?

4. Why is the term $e^{\gamma}$ introduced in Equation (11)? The purpose of this exponential parameterization is not clearly explained.

5. The paper states that the only requirement for $ G $ is that it be **invertible**. Since almost all random matrices are invertible in practice, it is unclear whether polar parameterization is strictly necessary. Would a simpler approach—such as adding a regularizer to prevent the smallest singular value from becoming too small—be sufficient?

---

> ### Author Rebuttal · Authors · 2026-03-31
>
> Q1:*Why Higher WikiText-2 PPL Does Not Necessarily Imply Lower 0-Shot Accuracy*
>
> A1: Thank you for pointing this out. We agree this trend can appear counterintuitive and believe this mainly comes from the calibration dataset.
>
> Our original TwinQuant results are calibrated on C4, while several baselines (e.g., SpinQuant and FlatQuant) use WikiText2 for both calibration and evaluation, which can make TwinQuant appear worse on WikiText2 PPL. We therefore added an additional experiment using WikiText2 as the calibration set for TwinQuant:
>
> |Model|Precision|Method|Avg.|Wiki2PPL|
> |---|---|---|---|---|
> |Qwen3-8B|W4A8|SpinQuant|69.9|13.5|
> |Qwen3-8B|W4A8|FlatQuant|69.9|14.0|
> |Qwen3-8B|W4A8|TwinQuant|**70.9**|**13.2**|
> |Qwen3-8B|W4A4|SpinQuant|68.3|14.8|
> |Qwen3-8B|W4A4|FlatQuant|69.3|14.4|
> |Qwen3-8B|W4A4|TwinQuant|**70.3**|**13.8**|
>
> Q2:*Why SpinQuant Is Not Included in the Throughput Comparison in Figure 5*
>
> A2: Thank you for pointing this out. Table 1 may suggest that SpinQuant should also appear in Figure 5, so we briefly clarify why it is excluded.
>
> - **Kernel support remains unavailable.** Although SpinQuant publicly releases its PTQ/rotation-learning code and an ExecuTorch export path, it does not provide an official real W4A4 kernel on NVIDIA GPUs. Its public deployment path is mainly described for W4A8, rather than a general-purpose W4A4 kernel.
>
> - **Accuracy is reproduced via fake quant.** Therefore, when reproducing SpinQuant, we could evaluate its accuracy only with fake quantization, where quantization is simulated but GEMM is still executed in 16-bit precision. This allows us to report perplexity in Table 1, but it does not reflect real low-bit execution efficiency.
>
> - **Runtime comparison would be unfair.** In contrast, the other methods in Figure 5 are paired with actual low-bit kernels, so including SpinQuant under a fake-quant backend would not be an *apples-to-apples* throughput comparison. For this reason, we did not include SpinQuant in Figure 5.
>
> Q3:*Clarifying the Meaning of “Overall” Perplexity in Line 346*
>
> A3: Thank you for pointing this out. Here, “overall” was intended as a qualitative cross-model observation under W4A8: TwinQuant delivers competitive WikiText2 perplexity across the evaluated models while remaining close to full precision, especially on larger Qwen3 models. We agree this wording is imprecise, since Table 1 reports only per-model WikiText2 perplexity and defines no aggregated perplexity metric. We will revise this statement accordingly.
>
> Q4:*Why the Exponential Parameterization Is Introduced in Equation (11)*
>
> A4: Thank you for pointing this out. We clarify it as follows:
>
> **Stable invertibility of $G$.**
> Equation (11) provides a stable parameterization of the layer-specific invertible transform $G$. Since our re-parameterized objective explicitly involves terms such as $Q^{-1}UG$ and $G^{-1}V$ (Sec. 4.1), $G$ must remain invertible throughout optimization; otherwise, its inverse would be undefined or numerically unstable.
>
> **Polar-form parameterization.**
> We therefore write $G$ in polar form as $G=PS$, and parameterize the symmetric factor as $S=e^{\gamma}(LL^{\top})$. This ensures that $S$ is positive definite, so $G$ remains nonsingular with a well-defined inverse (Sec. 4.2).
>
> **Practical benefits of parameterization.**
> This form separates the orthogonal component $P$ from the scaling/shearing component $S$, which is convenient for hybrid manifold optimization, and, together with the conditioning regularizer, keeps $G$ close to identity at initialization for stable training (Sec. 4.2).
>
> Q5:*Why Polar Parameterization Is Used Beyond Ensuring Invertibility*
>
> A5: Thank you for this insightful question. We would like to clarify that Eq. (11) is not introduced merely to ensure that $G$ is invertible.
>
> **More Than Invertibility.**
> In our method, $G$ is the layer-specific transform optimized on the general linear manifold. Its role is to provide extra scaling and shearing flexibility beyond orthogonal transforms, so Eq. (11) is part of the method design, not just a safeguard against singularity.
>
> **Aligned with Our Optimization.**
> In Sec. 4.1, $G$ appears directly in the low-rank re-parameterization through $U'=Q^{-1}UG$ and $V'=G^{-1}V$. In Sec. 4.2, we optimize it within our hybrid manifold framework. Under this setup, the polar form $G=PS$ gives a structured parameterization: $P$ handles the orthogonal part, while $S$ provides the extra non-orthogonal scaling/shearing freedom.
>
> **A Different Optimization Formulation.**
> We do not claim that polar parameterization is the only possible choice. Regularizing the smallest singular value could be an alternative, but it leads to a different formulation: an unconstrained parameterization with an added penalty term, rather than the structured, geometry-aware parameterization used in our hybrid manifold optimizer.

---

> > ### Author Rebuttal · Reviewer_Fq6M · 2026-04-02
> >
> > Thank you for the additional experiment in Q1. While calibrating TwinQuant on WikiText2 improves its WikiText2 PPL, the comparison still raises fairness concerns.
> >
> > Specifically, the original results use different calibration datasets across methods (C4 vs. WikiText2), which makes the comparison difficult to interpret. Performance differences may stem from dataset alignment rather than the method itself.
> >
> > For a fair comparison, it would be important to evaluate all methods under a consistent calibration setup (e.g., all on C4 or all on WikiText2). Could the authors clarify the reason to the choice of calibration data?

---

> > > ### Author Response · Authors · 2026-04-07
> > >
> > > Thank you very much again for your positive and constructive comments. For your further feedback, there may be a misunderstanding here: the additional experiment in Q1 is conducted under a consistent calibration setup (all on WikiText2) for all methods, including SpinQuant, FlatQuant and our TwinQuant. Therefore, the comparison in Q1 is fair. To avoid confusion, we will clarify this point in the revision and revise the experimental setup accordingly.

---

### Official Review · Reviewer_VbsX · 2026-03-10

**Soundness:** 3
**Presentation:** 3
**Significance:** 3
**Originality:** 2
**Overall Recommendation:** 4
**Confidence:** 4

**Summary:**

This paper introduces TwinQuant a quantization framework based on invertible linear transformations and low-rank decompositions of linear layers. In contrast to previous work, TwinQuant proposes a solution in which both the low-rank and resisual weight components are quantized to allow for higher residual rank. The effectiveness of the proposed method is empirically validated both in terms of accuracy and throughput of quantized LLM models.

**Compliance With Llm Reviewing Policy:**

Affirmed.

**Final Justification:**

The authors addressed my concerns by including the crucial comparison with SVDQuant in their response. Hence my main concerns have been addressed.

**Key Questions For Authors:**

1. **SVD‑Quant comparison:**
Can you provide a direct comparison with SVD‑Quant (same model, datasets, hardware, and sequence lengths), including a few points on the Pareto curve of accuracy vs. Latency/throughput/memory for differen ranks?

2. **QuaRot throughput:**
What exact benchmark setup led to QuaRot having lower throughput than a 16‑bit model? Please include a breakdown (rotation cost, matmuls, quant/dequant, memory) and whether FHT was used.

3. **Ablation interpretation:**
In Sec. 5.4, the authors mention “replacing learnable transforms with fixed Hadamard improves performance,” which transforms are replaced, and what does “+Low‑Rank” mean exactly (is it SVD‑Quant‑style without learned $G$)?

**Limitations:**

The authors adequately discussed the limitations.

**Strengths And Weaknesses:**

# Strengths

* **Clarity and accessibility:** The method is straightforward and well‑explained. The paper is easy to follow and motivates the decomposition‑plus‑quantization design clearly.

* **Convincing evaluation on core metrics:** The method outperforms existing 4‑bit baselines on perplexity and common‑sense reasoning, with consistent gains observed across LLama and Qwen models.

* **Systems angle:** The custom kernel design supports end‑to‑end evaluation (accuracy and throughput), helping connect algorithmic choices to practical deployment.

# Weaknesses

While the approach is clearly presented and empirically promising, the lack of a direct accuracy and compute/latency comparison with SVD‑Quant makes it difficult to assess the contribution and practical trade‑offs. I am willing to increase my score whenever this primary concern is adressed


1. **Missing direct comparison with SVD‑Quant (accuracy and compute):**
Given the conceptual proximity—both methods preserve signal via a low‑rank (or side) path while quantizing the main path—the absence of a head‑to‑head comparison with SVD‑Quant is the most significant gap. Equation 3 reports a solution in which activations are quantized in the low-rank path. This is inconsistent with what reported in the SVDQuant paper (Eqution 4, Figure 6b) in which the low-rank path uses activations in high precision (before quantization). This difference is quite crucial since the biggest challenge for W4A4 quantization is usually the activation quantization part. Without this, it is difficult to judge novelty and practical benefit.

2. **QuaRot throughput benchmarking (Figure 5) appears counterintuitive:**
QuaRot applies one online rotation per layer (before the down‑projection) and can leverage the Fast Hadamard Transform (FHT). Under typical settings, this should be very cheap relative to 16‑bit inference. It is therefore surprising that QuaRot’s throughput is reported substantially lower than a 16‑bit model. A more detailed performance breakdown could help the reader to interpret the results.


## Minor Issues
* $E_x$, $E_R$, $E_{W_UV}$ and some of the other components in equation 5 are not clearly defined

* $G$ referred to as “rotation” in Figure 3 even if it consists of a generic invertible matrix (not necessarily orthogonal).

* The experimental setting for the study reported in section 5.4 is not fully clear (See questions)

---

> ### Author Rebuttal · Authors · 2026-03-31
>
> Q1:*Direct Comparison with SVDQuant Across Different Ranks.*
>
> A1: We have added a direct comparison with SVDQuant under the same setup on LLaMA3-8B with an RTX4090 GPU. Accuracy is the average over the six zero-shot commonsense reasoning tasks used in the paper, while throughput and memory overhead are measured at input/output lengths 512/512 with batch size=8. We evaluate both methods across multiple ranks and summarize the results below:
>
> |**Method**|**Rank**|**Avg.Accuracy(%)**|**Throughput**|**MemoryOverhead(MB)**|
> |---|---:|---:|---:|---:|
> |SVDQuant|32|68.2|833.5|3.51|
> |SVDQuant|64|69.5|811.0|3.66|
> |SVDQuant|128|70.5|789.5|3.98|
> |SVDQuant|256|70.6|709.2|4.60|
> |TwinQuant|32|68.0|861.4|3.39|
> |TwinQuant|64|69.7|843.6|3.43|
> |TwinQuant|128|70.8|816.5|3.51|
> |TwinQuant|256|70.9|758.3|3.67|
>
> Several insights can be drawn from the table.
>
> **Rank–accuracy–overhead trade-off.** As rank increases, both methods achieve higher zero-shot accuracy, but with lower throughput and higher memory overhead, confirming the expected trade-off.
>
> **Rank 128 is the sweet spot.** For TwinQuant, rank 128 provides the best practical balance because it captures most of the accuracy gain while avoiding the larger throughput drop and memory increase.
>
> **High-precision low-rank branch hurts efficiency.** At the same rank, SVDQuant has lower throughput since it keeps the low-rank branch in 16-bit, increasing both computation and memory traffic compared with TwinQuant’s fully 4-bit design.
>
> Q2:*Why QuaRot Has Lower Throughput than the FP16 Baseline and the Role of FHT?*
>
> A2: Thanks for your question. We clarify the QuaRot baseline setting as follows:
>
> **Original setup without FHT.** The QuaRot throughput reported in Fig.5 is measured *without FHT* for online rotation. To make this explicit, we further benchmark an FHT-enabled QuaRot version under the same setting. The results below show consistent throughput gains from FHT in end-to-end, prefill, and decoding.
>
> |Method|FHT|bz=1|bz=2|bz=4|bz=8|
> |---|---:|---:|---:|---:|---:|
> |QuaRot(E2E)|No|23.3|46.4|91.9|195.4|
> |QuaRot(E2E)|Yes|35.2|70.5|141.8|282.8|
> |QuaRot(Prefill)|No|4831.8|7646.3|7689.6|7685.5|
> |QuaRot(Prefill)|Yes|7141.5|11241.9|13086.6|13078.8|
> |QuaRot(Decoding)|No|12.5|23.8|45.3|96.3|
> |QuaRot(Decoding)|Yes|13.4|34.8|70.9|147.2|
>
> **Decoding is the main bottleneck.** Even with FHT, QuaRot still does not surpass the FP16 TensorRT-LLM baseline in our setting. The key reason is decoding: prefill throughput is already high, while decoding remains much slower. Since decoding generates tokens autoregressively, the overhead of online transforms, quant/dequant, packing, and extra kernel launches is incurred at every step, making it much harder to amortize than in prefill.
>
> **Breakdown clarifies the bottleneck shift.** We further add a latency breakdown. After enabling FHT, the rotation cost drops substantially (from 38.3% to 6.8%), and end-to-end latency is correspondingly reduced. The dominant cost then shifts to matmuls and other low-bit runtime overheads, including quant/dequant, packing, and memory/data movement. This shows that FHT effectively alleviates rotation overhead, while the remaining decoding-time low-bit execution overhead still limits QuaRot’s end-to-end throughput. We will include both the no-FHT and FHT-enabled results, together with this breakdown, in the revision.
>
> |Method|FHT|Rotation/FHT(ms,%)|Matmuls(ms,%)|Quant/Dequant+Pack/Unpack(ms,%)|Memory/DataMovement(ms,%)|Other(ms,%)|Total(ms)|
> |---|---:|---:|---:|---:|---:|---:|---:|
> |QuaRot(E2E)|No|16835.7(38.3%)|12826.2(29.2%)|5721.4(13.0%)|4943.3(11.2%)|3621.9(8.2%)|43948.5|
> |QuaRot(E2E)|Yes|1978.2(6.8%)|12829.2(44.1%)|5701.8(19.6%)|4945.4(17%)|3636.4(12.5%)|29090.9|
>
> Q3:*Clarifying the Ablation Settings in Sec. 5.4.*
>
> A3: Thank you for your question. We first clarify that, relative to TwinQuant, “+Hadamard” removes the learnable transforms and uses fixed Hadamard transforms instead. The ablation settings are:
>
> **Naive 4-bits** means directly quantizing the original weight to W4A4 without decomposition.
>
> **+Low-Rank** adds only the SVD-based low-rank decomposition, where the weight is split into low-rank and residual branches and both branches are quantized to 4 bits; this is the decomposition-only baseline.
>
> **+Hadamard** keeps the same decomposed structure but replaces all learnable transforms (e.g., the global orthogonal transforms and layer-specific transforms) with fixed Hadamard transforms, to test whether fixed orthogonal mixing alone is sufficient.
>
> **TwinQuant** replaces these fixed transforms with the learnable global orthogonal transforms and layer-specific invertible transforms introduced in Sec.4.1.
>
> Under this interpretation, Table 2 isolates the gains from (i) decomposition itself, (ii) fixed orthogonal smoothing, and (iii) learnable component-specific transforms. We will clarify these definitions in Sec.5.4 and in the caption of Table 2.

---

> > ### Author Rebuttal · Reviewer_VbsX · 2026-04-02
> >
> > I wish to thank the authors for their convincing and exhaustive responses. I strongly recommend including the SVDQuant comparison in the main text as it strengthen the value proposition for the proposed TwinQuant method.
> > Please modify the text prior to equation 3 to clarify that SVD quant uses a high-precision low rank path while TwinQuant uses quantized activations in both paths.
> >
> > Since my main concerns have been addressed, I will increase my score accordingly.

---

> > > ### Author Response · Authors · 2026-04-07
> > >
> > > Thank you very much for your positive and constructive feedback. We are glad that our responses have addressed your main concerns. Following your suggestion, we will include the SVDQuant comparison in the main text to further strengthen the value proposition of TwinQuant. We will also revise the text before Eq. (3) to clarify that SVDQuant uses a high-precision low-rank path, while TwinQuant uses quantized activations in both paths. We sincerely appreciate your helpful suggestions and support.

---

### Official Review · Reviewer_4r1g · 2026-03-11

**Soundness:** 4
**Presentation:** 3
**Significance:** 4
**Originality:** 4
**Overall Recommendation:** 5
**Confidence:** 5

**Summary:**

The paper introduces a quantization approach for LLMs that combines ideas from low-rank decomposition and learnable transformations to improve accuracy under 4-bit weight and activation quantization.

Prior work (SVDQuant) on diffusion models decomposes weight matrices into a low-rank component and a residual, keeping the low-rank component in higher precision (BF16) while quantizing the residual. The authors show that this approach does not transfer well to LLMs, as achieving low quantization error in the residual requires relatively large ranks. To address this limitation, the proposed method also quantizes the low-rank component while applying learnable transformations, similar to those used in SpinQuant, to reduce quantization error.

In addition, the method leverages the structure of the low-rank decomposition to introduce layer-specific learnable transformations, which contribute significantly to reduce the overall quantization error. The resulting algorithm, called TwinQuant, is evaluated on Llama-3 and Qwen3 models and compared against several recent weight and activation quantization methods, including QuaRot, SpinQuant, and FlatQuant. The results show improved accuracy relative to these baselines under comparable quantization settings.

Finally, the authors implement specialized kernels that fuse the inference operations for the quantized low-rank and residual components, aiming to reduce memory and latency overhead during inference.

**Compliance With Llm Reviewing Policy:**

Affirmed.

**Final Justification:**

This is a well written paper that introduced new techniques to address challenges in LLM compression. The authors have been attentive to minor presentation issues I raised in the rebuttal and I am confident in my recommendation for acceptance.

**Key Questions For Authors:**

No questions

**Limitations:**

Yes.

**Strengths And Weaknesses:**

**Soundness**

The paper provides a careful analysis of the sources of quantization error when applying low-rank decomposition in the context of weight and activation quantization. In particular, the authors analyze the limitations of applying the SVDQuant decomposition strategy directly to LLMs and motivate the need to quantize both the low-rank and residual components. The paper also provides theoretical intuition for the use of transformation operators to reduce quantization error.

The experimental evaluation is comprehensive and includes comparisons against several recent state-of-the-art quantization approaches. Experiments are conducted on multiple model families and demonstrate consistent improvements in accuracy under comparable quantization settings. Overall, the methodology appears technically sound and the empirical evidence supports the main claims of the paper.

**Presentation**

The paper is clearly written and well structured, and the overall narrative is easy to follow. The related work section is comprehensive and positions the proposed method well within the existing literature.

I have only minor suggestions to further improve clarity:

1. The authors briefly introduce quantization concepts but do not explain group quantization, which is used throughout the paper. A short explanation would improve completeness for readers who may not be familiar with this technique.

2. The appendix contains useful analyses regarding computational cost, optimization setup, and sensitivity to the chosen rank. It would be helpful for the authors to briefly mention these analyses in the main text and refer the reader to the appendix for details.

3. In Figure 5, it is not clear how TensorRT-LLM fits into the picture. Does this represent BF16 execution?

**Significance**

Efficient inference of large language models remains an important practical challenge, and quantization is one of the most widely used approaches for reducing memory footprint and computational cost. In particular, achieving accurate 4-bit weight and activation quantization is an active research topic.

The paper proposes a method that improves accuracy under this quantization regime by combining low-rank decomposition with learnable transformation operators. A notable aspect of the approach is the introduction of layer-specific learnable transformations that exploit the structure of the low-rank decomposition, which helps reduce quantization error and contributes to the improvements observed in the experiments.

In addition, the authors implement specialized kernels that fuse the inference operations associated with the quantized low-rank and residual components. This systems contribution helps ensure that the proposed method can be deployed with limited memory and latency overhead.

Overall, the work addresses a relevant problem for efficient LLM deployment and may provide a useful approach for improving accuracy under aggressive quantization settings.

**Originality**

The paper combines ideas from low-rank decomposition and learnable transformation-based quantization methods. While these components have appeared in prior work (e.g., SVDQuant and SpinQuant), the paper proposes a formulation that jointly quantizes the low-rank and residual components and introduces layer-specific transformation operators that leverage the decomposition structure.

This combination of techniques, together with the proposed optimization procedure, represents a novel approach to 4-bit LLM quantization and is clearly distinguished from closely related prior work.

---

> ### Author Rebuttal · Authors · 2026-03-31
>
> Q1:*Definition of Group-Wise Quantization.*
>
> A1: Thanks for your suggestion.
>
> **Define group-wise quantization.** Thank you for this helpful suggestion. Group-wise quantization partitions each weight or activation tensor into fixed-size groups, and assigns an individual quantization scale to each group, rather than sharing one global scale across the entire tensor.
>
> **Add this definition in Sec. 5.1.** In the revision, we will add this definition in Sec. 5.1 and state explicitly that all experiments use a group size of 128 for both weights and activations.
>
> Q2:*Mentioning Rank-Sensitivity Analysis in the Main Text.*
>
> A2: Thank you for this helpful suggestion. These analyses are currently presented in the appendix and are not explicitly mentioned in the main text. In the revision, we will briefly mention the computational-cost, optimization-setup, and rank-sensitivity results at the relevant points in the main text,  summarize the key takeaway for the default rank $r=128$.
>
> Q3:*Clarifying the TensorRT-LLM Baseline in Figure 5.*
>
> A3: Thank you for pointing this out. Yes, in Fig. 5, TensorRT-LLM refers to our FP16 TensorRT-LLM baseline (not BF16). We agree that this should be stated more clearly. In the revision, we will clarify this explicitly in the figure caption and the corresponding text in Sec. 5.3.

---

> > ### Author Rebuttal · Reviewer_4r1g · 2026-03-31
> >
> > As I mentioned in my original review, I believe the paper is well structured and introduces a significant contribution to the field. I only had minor presentation issues. The authors acknowledged the in their rebuttal and proposed to address them accordingly in the final version.

---

> > > ### Author Response · Authors · 2026-04-07
> > >
> > > Thank you very much for your recognition and support. We are glad that you appreciate the structure and contribution of our paper. We will revise the presentation issues you pointed out in the final version, as described in our rebuttal. Thank you again for your valuable feedback.

---

### Official Review · Reviewer_WWnh · 2026-03-12

**Soundness:** 2
**Presentation:** 1
**Significance:** 2
**Originality:** 2
**Overall Recommendation:** 4
**Confidence:** 4

**Summary:**

This paper introduces TwinQuant, a novel quantization framework that learns component-specific transformations via joint optimization over the Stiefel and general linear manifolds, aiming to flatten distributions and reduce dynamic-range imbalance. The authors first identify a limitation of existing SVD-based two-component quantization methods: in mainstream LLMs, small ranks leave large residuals that suffer sharp 4-bit accuracy loss due to slow spectral decay, while large ranks are impractical because the low-rank branch is typically kept in high precision and becomes prohibitively expensive. To address these challenges, the paper proposes TwinQuant, which applies a global rotation matrix and a layer-specific transformation, and optimizes them jointly on a small calibration set to enable quantization-aware learnable subspace decomposition. Furthermore, the authors introduce a fused dual-component kernel that enables efficient end-to-end execution. TwinQuant achieves strong accuracy among popular PTQ methods and reports up to 2.11× end-to-end speedup over an FP16 baseline.

**Compliance With Llm Reviewing Policy:**

Affirmed.

**Final Justification:**

My concerns have been adequately addressed.

**Key Questions For Authors:**

1. The method is motivated by orthogonal rotations redistributing outliers, but the paper claims learnable subspace decomposition. Could the authors clarify how the proposed transforms actually learn the subspace?

2. Could the authors explain why the proposed method yields higher WikiText perplexity than other baselines in many cases (e.g., W4A8 on Qwen3-4B and LLaMA3-8B, and W4A4 on LLaMA3-8B) and what factors might be driving this behavior?

**Limitations:**

No. The paper should discuss the limitations of the proposed work.

**Strengths And Weaknesses:**

### Strengths
- The reported improvements appear more pronounced as model size increases, consistent with the fact that a fixed low-rank setting becomes relatively cheaper and the proposed reshaping is more beneficial at scale.
- The paper does not stop at a mathematical formulation and algorithm. It also provides a fused dual-component kernel that addresses the practical latency bottlenecks of two-stage low-rank execution under 4-bit TensorCore constraints.
- The hybrid manifold optimization which applies Stiefel for global rotations and GL for layer-specific transforms is a novel way to respect structural constraints while increasing expressivity where it matters.


### Weaknesses

- In the Introduction, the paper argues that low-rank decomposition is less effective for mainstream LLMs due to slow rank decay, but Section 3.2 shifts the focus to skewed 4-bit quantization error in the low-rank branch, and the link between these issues is not made explicit. Moreover, the experiments largely fix the rank (e.g., r=128) rather than analyzing the rank–accuracy and overhead trade-off, making it unclear how directly TwinQuant addresses the slow-rank-decay limitation in practice.

- The optimization objective ultimately minimizes reconstruction error on a calibration set, but the paper does not sufficiently justify why reconstruction error is the right surrogate for the specific quantization failure mode highlighted earlier. A more explicit causal chain from slow spectral decay, to the need for higher ranks, to the distributional pathologies of the low-rank factors under 4-bit quantization, and finally to why reconstruction error minimization addresses these pathologies would strengthen the overall logical flow.

- The paper motivates transformation matrices by arguing that orthogonal rotations can redistribute outliers across groups and reduce 4-bit distortion. However, it is not fully clear how this rotation-based intuition directly supports the claimed learnable subspace decomposition. The former reads as distribution mixing and flattening, while the latter suggests learning the decomposition itself. A clearer explanation of how the proposed re-parameterization and hybrid-manifold optimization correspond to learning the subspace, rather than simply applying quantization-friendly preconditioners, would strengthen the narrative

---

> ### Author Rebuttal · Authors · 2026-03-31
>
> Q1:*Clarifying the Link Between Slow Rank Decay and Low-Rank Quantization Error*
>
> A1: Thank you for pointing this out. We clarify the link as follows:
>
> **Slow rank decay in LLMs**
> As noted in the Introduction, mainstream LLMs exhibit much slower singular-value decay (Fig. 1a). Under the small-rank setting used in prior SVD-based methods (e.g., rank=32 in SVDQuant), the residual branch remains large and incurs substantial 4-bit quantization error (Fig. 1b), indicating that a higher rank is needed.
>
> **High rank is needed but costly**
> However, a higher rank also increases memory and computation overhead, since prior SVD-based methods keep the low-rank branch in high precision.
>
> **Directly quantizing the low-rank branch leads to large error.**
> Quantizing the low-rank branch is therefore a natural way to recover efficiency, but in mainstream LLMs this branch is often scale-imbalanced and heavy-tailed, making direct 4-bit quantization highly error-prone (Fig. 2).
>
> **TwinQuant learns quantization-friendly decomposition.**
> SVDQuant relies on a high-precision low-rank branch to absorb outliers, which is not well suited to the setting where both branches are quantized. TwinQuant instead learns a decomposition in which both the low-rank and residual branches are more quantization-friendly.
>
> Q2:*Analysis of Rank–Accuracy and Overhead Trade-off*
>
> A2: Thank you for raising this point. The trade-off is discussed in Appendix A.5.
>
> Q3:*Why Reconstruction Error Is the Right Surrogate for the Quantization Failure Mode*
>
> The causal chain is: slow spectral decay in mainstream LLMs makes higher-rank decomposition necessary, and 4-bit quantization of the low-rank branch then becomes the next bottleneck.
>
> However, our reconstruction error is not intended to surrogate the quantization error of the low-rank branch alone.
>
> **Why is the low-rank branch error not the right target?**
> The low-rank branch error alone is not the right target, because the final layer output is jointly determined by activation quantization, the low-rank branch, and the residual branch. Therefore, a small low-rank-only error does not necessarily imply a small final output error.
>
> **Why is reconstruction error the right surrogate for this failure mode?**
> In Eq. (4), reconstruction error is the discrepancy between the quantized and full-precision layer outputs on the calibration set. It therefore reflects the actual quantization error in the final computation, rather than the low-rank branch alone. Eq. (5) further shows that this discrepancy is jointly determined by the activation, low-rank, and residual terms.
>
> Q4:*From Rotation Intuition to Learnable Subspace Decomposition*
>
> A4: Thank you for this insightful comment. Orthogonal rotation is only the starting intuition of our method; the key contribution of TwinQuant is to make the decomposition itself learnable under the quantization objective. We clarify this connection from three perspectives.
>
> **The objective is quantization-aware.**
> SVDQuant first fixes a truncated SVD decomposition and then quantizes the resulting branches, so the decomposition is not learned under the final post-quantization layer-output error. In contrast, TwinQuant directly optimizes that error over the activation, low-rank, and residual terms, making the decomposition itself quantization-friendly.
>
> **Re-parameterization makes the decomposition variable.**
> Its role is not to precondition a fixed decomposition, but to make the decomposition itself variable under the same rank budget. In Eq. (6), TwinQuant preserves the full-precision mapping while changing the representation seen by the quantizers. Different $Q$ and $G$ therefore induce different low-rank/residual allocations under the same rank constraint.
>
> **Hybrid-manifold optimization makes it learnable.**
> This decomposition becomes learnable because $Q$ and $G$ are optimized under the quantization objective rather than fixed. $Q$ reshapes the activation and residual terms, while $G$ adapts the low-rank branch. Optimizing them on the Stiefel and general linear manifolds lets TwinQuant learn the low-rank/residual structure itself.
>
> Q5:*Why TwinQuant Shows Higher WikiText-2 Perplexity in Some Cases*
>
> A5: Thank you for pointing this out. A key factor is the calibration–evaluation mismatch: our original TwinQuant results were calibrated on C4 but evaluated on WikiText-2, while several baselines (e.g., SpinQuant/FlatQuant) use WikiText-2 for both. We therefore added a new setting where TwinQuant is also calibrated and evaluated on WikiText-2:
>
> |Model|Precision|Method|Wiki2PPL|
> |---|---|---|---:|
> |Qwen3-4B|W4A8|SpinQuant|19.8|
> |Qwen3-4B|W4A8|FlatQuant|19.4|
> |Qwen3-4B|W4A8|TwinQuant|18.3|
> |Qwen3-4B|W4A4|SpinQuant|21.7|
> |Qwen3-4B|W4A4|FlatQuant|20.0|
> |Qwen3-4B|W4A4|TwinQuant|18.6|
> |LLaMA3-8B|W4A8|SpinQuant|9.0|
> |LLaMA3-8B|W4A8|FlatQuant|9.8|
> |LLaMA3-8B|W4A8|TwinQuant|8.9|
> |LLaMA3-8B|W4A4|SpinQuant|9.5|
> |LLaMA3-8B|W4A4|FlatQuant|11.0|
> |LLaMA3-8B|W4A4|TwinQuant|9.4|

---

> > ### Author Rebuttal · Reviewer_WWnh · 2026-04-03
> >
> > Thank you for the response. My concerns have been adequately addressed. I will increase my score.

---

> > > ### Author Response · Authors · 2026-04-07
> > >
> > > Thank you very much for your recognition. We are glad that the concerns have been resolved, and we sincerely appreciate your increased score. Thank you again for your support.

---

### Decision · Program_Chairs · 2026-04-30

**Decision:**

Accept (regular)

**Comment:**

The paper proposes TwinQuant, a 4-bit quantization framework for LLMs that combines low-rank decomposition with learnable transformations applied to both the low-rank and residual weight components. It does so via a joint optimization on Stiefel and general linear manifolds to reduce dynamic range imbalance and improve quantization accuracy. The paper also introduces a fused kernel to address a system bottleneck in executing the low-rank component efficiently under full 4-bit quantization. Empirical results demonstrate improved accuracy over prior PTQ methods and competitive throughput, highlighting the effectiveness of the learned subspace transformations and system-level design.

Reviewers appreciated the paper for its clear methodology, use of low‑rank and manifold‑based optimization, and strong empirical results with practical applicability supported by an efficient fused kernel implementation. Remaining issues were adequately addressed during the rebuttals. The authors are advised to include the additional elements and clarifications that came up during the discussion in the full manuscript to improve the presentation and clarity.